# A multi-classifier system integrated by clinico-histology-genomic analysis for predicting recurrence of papillary renal cell carcinoma

Kang-Bo Huang [1,2,3,17], Cheng-Peng Gui[1,17], Yun-Ze Xu[4,17], Xue-Song Li[5,17], Hong-Wei Zhao[6,17], Jia-Zheng Cao[7,17], Yu-Hang Chen[1], Yi-Hui Pan[8], Bing Liao[9], Yun Cao [3,10], Xin-Ke Zhang [3,10], Hui Han[2,3], Fang-Jian Zhou [2,3], Ran-Yi Liu [3], Wen-Fang Chen[9], Ze-Ying Jiang[9], Zi-Hao Feng[1], Fu-Neng Jiang[11], Yan-Fei Yu[5], Sheng-Wei Xiong[5], Guan-Peng Han[5], Qi Tang[5], Kui Ouyang[6], Gui-Mei Qu[12], Ji-Tao Wu[6], Ming Cao[4], Bai-Jun Dong [4], Yi-Ran Huang[4], Jin Zhang[4], Cai-Xia Li[13], Pei-Xing Li[13], Wei Chen[1], Wei-De Zhong [11], Jian-Ping Guo [14], Zhi-Ping Liu [15], Jer-Tsong Hsieh [16], Dan Xie [3,10], Mu-Yan Cai [3,10], Wei Xue [4] ✉, Jin-Huan Wei[1] ✉ & Jun-Hang Luo [1,14] ✉

Integrating genomics and histology for cancer prognosis demonstrates promise. Here, we develop a multi-classifier system integrating a lncRNA-based classifier, a deep learning whole-slide-image-based classifier, and a clinicopathological classifier to accurately predict post-surgery localized (stage I–III) papillary renal cell carcinoma (pRCC) recurrence. The multi-classifier system demonstrates significantly higher predictive accuracy for recurrence-free survival (RFS) compared to the three single classifiers alone in the training set and in both validation sets (C-index 0.831-0.858 vs. 0.642-0.777, p < 0.05). The RFS in our multi-classifier-defined high-risk stage I/II and grade 1/2 groups is significantly worse than in the low-risk stage III and grade 3/4 groups (p < 0.05). Our multi-classifier system is a practical and reliable predictor for recurrence of localized pRCC after surgery that can be used with the current staging system to more accurately predict disease course and inform strategies for individualized adjuvant therapy.

In 2022, kidney cancer represented 4.2% of all new cancer cases, and its incidence has been increasing[1]. Over 90% of kidney cancer cases are renal cell carcinoma (RCC), which includes various subtypes with distinct histologic features, clinical courses, and responses to therapy[2–4]. Papillary renal cell carcinoma (pRCC) is the second most common histological subtype of RCC, accounting for approximately 20% of all cases[5,6]. Recently, a phase 3 trial (EVEREST) revealed that RCC patients (including pRCC) in the very-high-risk group could benefit from adjuvant therapy[7]. Current staging system is insufficient for more accurate risk stratification. There is an urgent need for a more comprehensive classifier to accurately predict post-surgery recurrence.

Approaches to molecularly classify RCC, when considered along with clinicopathological risk factors, can more accurately predict disease course and support physicians in devising more informed, customized treatment decisions and follow-up plans. Several validated prognostic genetic signatures for RCC have been developed, such as the 16-gene

assay[8], Clearcode34[9], and the six-single-nucleotide polymorphism (SNP)-based classifier[10]; however, most of these molecular classifiers were constructed based on clear cell renal cell carcinoma (ccRCC), and they cannot be applied to patients with pRCC. Recently, several classifiers for pRCC have been developed using genome-wide DNA methylation, messenger RNA (mRNA), and microRNA profiling with The Cancer Genome Atlas (TCGA) dataset. These studies suggested that pRCC with CpG island methylator phenotype (CIMP) is associated with particularly poor survival[6,11]. However, the predictive accuracy of these classifiers in pRCC remains unsatisfactory, thereby limiting their clinical application.

Long non-coding RNAs (lncRNAs) are transcripts longer than 200 nucleotides that lack protein-coding potential[12], which are involved in multi-level critical regulation of biological processes[13-15]. Various lncRNA transcripts are considered biomarkers that can effectively predict the clinical outcome of many cancers[16-19]. Another recently developed approach to characterize tumors is the use of deep learning to identify histopathologic and molecular features on H&E-stained tumor slides[20-23]. Indeed, a whole-slide-image (WSI)-based deep learning model has been shown to accurately predict outcomes of patients with colorectal cancer and soft tissue sarcoma[20,24]. In this study, we developed a multi-classifier system based on genome-wide lncRNA profiling, histopathologic images, and clinicopathological factors that improves risk stratification for patients with pRCC. We validated the predictive accuracy and reproducibility of the multi-classifier system in two independent cohorts, including cases from multiple centers in China and the TCGA dataset.

## Results

### Constructing a lncRNA-based classifier in the training set

We included in our study 793 patients with pRCC from three independent cohorts: the training set, the independent validation set, and the TCGA set (Supplementary Fig. 1). Clinical features of patients in the three cohorts are described in Table 1. To develop a lncRNA-based classifier for predicting tumor recurrence, we first analyzed 53 paired fresh-frozen pRCC and adjacent normal tissues by RNA-seq in the discovery set and looked for differentially expressed lncRNAs in pRCC tumors compared to normal tissue across the genome (Fig. 1A). Based on the genome-wide analysis of lncRNAs, 40 lncRNAs were identified as significantly differentially expressed genes (Fig. 1A; Supplementary Table 1). The heat map clearly distinguishes between levels of these 40 lncRNA in tumors versus adjacent normal tissues (Fig. 1A).

We performed quantitative real-time polymerase chain reaction (qRT–PCR) to quantify the expression of the 40 differentially expressed lncRNAs by using formalin-fixed, paraffin-embedded (FFPE) tissue samples from the training set of 382 patients. The training set included 189 patients from South China (two centers) and 193 patients from East China (one centers). Then we conducted univariate Cox regression analysis of recurrence-free survival (RFS) based on each of the 40 lncRNAs (Supplementary Table 2). We then used a multivariate LASSO Cox regression model to select four lncRNAs to generate a lncRNA-based risk score for RFS for each patient (Fig. 1A), using the following formula:

$$\begin{aligned} \text{Four} - \text{lncRNA} - \text{based risk score} = &(0.4537 \times \text{AC099850.3}) \\ &+ (0.8549 \times \text{lnc} - \text{TRDMT1} - 5) \\ &+ (0.4143 \times \text{CYTOR}) + (1.2739 \times \text{LUCAT1}). \end{aligned} \quad (1)$$

The expression of the four lncRNAs is shown in Supplementary Fig. 2. We assessed the association between lncRNA-based risk score and disease recurrence status, and we observed that patients with higher risk scores generally had higher recurrence rates than patients with lower risk scores (Supplementary Fig. 3).

### Constructing a WSI-based classifier in the training set

To build a WSI-based classifier, we chose a total of 182 patients with pRCC from the training set who had either a distinct good or poor

**Table 1 | Baseline characteristics of patients in the three sets**

| | Training set (n = 382) | Independent validation set (n = 207) | TCGA set (n = 204) | Total (n = 793) |
|---|---|---|---|---|
| Sex | | | | |
| Woman | 83 (21.7%) | 48 (23.2%) | 50 (24.5%) | 181 (77.2%) |
| Man | 299 (78.3%) | 159 (76.8%) | 154 (75.5%) | 612 (22.8%) |
| Age, years | | | | |
| <60 | 223 (58.4%) | 112 (54.1%) | 86 (42.2%) | 421 (53.1%) |
| ≥60 | 159 (41.6%) | 95 (45.9%) | 118 (57.8%) | 372 (46.9%) |
| WHO/ISUP grade | | | | |
| 1 | 11 (2.9%) | 11 (5.3%) | 12 (5.9%) | 34 (4.3%) |
| 2 | 216 (56.5%) | 113 (54.6%) | 108 (52.9%) | 437 (55.1%) |
| 3 | 145 (38.0%) | 79 (38.2%) | 81 (39.7%) | 305 (38.5%) |
| 4 | 10 (2.6%) | 4 (1.9%) | 3 (1.5%) | 17 (2.1%) |
| Pathologic stage[a] | | | | |
| I | 241 (63.1%) | 155 (74.9%) | 138 (67.6%) | 534 (67.3%) |
| II | 56 (14.7%) | 20 (9.7%) | 20 (9.8%) | 96 (12.1%) |
| III | 85 (22.2%) | 32 (15.4%) | 46 (22.6%) | 163 (20.6%) |
| Recurrence-free survival (%, 95% CI) | | | | |
| 3 years | 88.1 (86.4–89.8) | 90.6 (88.6–92.6) | 80.1 (76.6–83.6) | 87.1 (85.8–88.4) |
| 5 years | 82.8 (80.8–84.8) | 82.6 (79.8–85.2) | 74.7 (70.3–79.1) | 81.0 (79.4–82.6) |
| 7 years | 80.4 (78.2–82.6) | 76.7 (73.3–80.1) | 64.5 (57.8–71.2) | 76.8 (75.0–78.6) |
| Overall survival (%, 95% CI) | | | | |
| 3 years | 90.2 (88.7–91.7) | 91.7 (89.8–93.6) | 88.7 (85.8–91.6) | 90.3 (89.2–91.4) |
| 5 years | 84.8 (82.9–86.7) | 85.6 (83.0–88.2) | 78.2 (73.4–83.0) | 84.2 (82.7–85.7) |
| 7 years | 80.4 (78.1–82.7) | 80.1 (76.9–84.3) | 78.2 (73.4–83.0) | 79.8 (78.1–81.5) |

*WHO* World Health Organization, *ISUP* International Society of Urological Pathology.
[a]Pathologic stage was classified according to the TNM 2017 staging system. Source data are provided as a Source Data file.

outcome as a development set. Patient who had more than 7 years of follow-up after surgery and had no record of recurrence was assigned to the distinct good outcome group (n = 127). The distinct poor outcome group consisted of patients with record of recurrence fewer than 3 years after surgery (n = 55). We then used the representative H&E-stained FFPE tumor tissue sections of each patient to scan their digital WSI, and applied deep learning to create the WSI-based classifier for predicting recurrence in patients with pRCC (Fig. 1B; see Methods). The predictive accuracy of the WSI-based score with the 10× WSI (C-index 0.723, 95% CI 0.553–0.994) was higher than that with the 40× WSI (C-index 0.675, 95% CI 0.514–0.727) (Supplementary Fig. 4). We calculated the WSI-based risk score (with 10× resolution) for each patient and found that patients with higher WSI-based risk scores also had higher recurrence rates (Supplementary Fig. 5).

### Constructing a clinicopathological classifier in the training set

Meanwhile, we evaluated several clinicopathological factors including age, sex, grade and pathologic stage in univariate and multivariate Cox regression analyses, and found that grade and pathologic stage were significant factors for predicting RFS in the training set. Further multivariate analysis showed that grade and pathologic stage were also independent prognostic factors after adjusting for age and sex (Supplementary Table 3). Thus, we developed a clinicopathological classifier using Cox regression based on the two predictors:

$$\text{clinicopathological risk score} = 0.5670 \times \text{grade} + 0.5326 \times \text{stage} \quad (2)$$

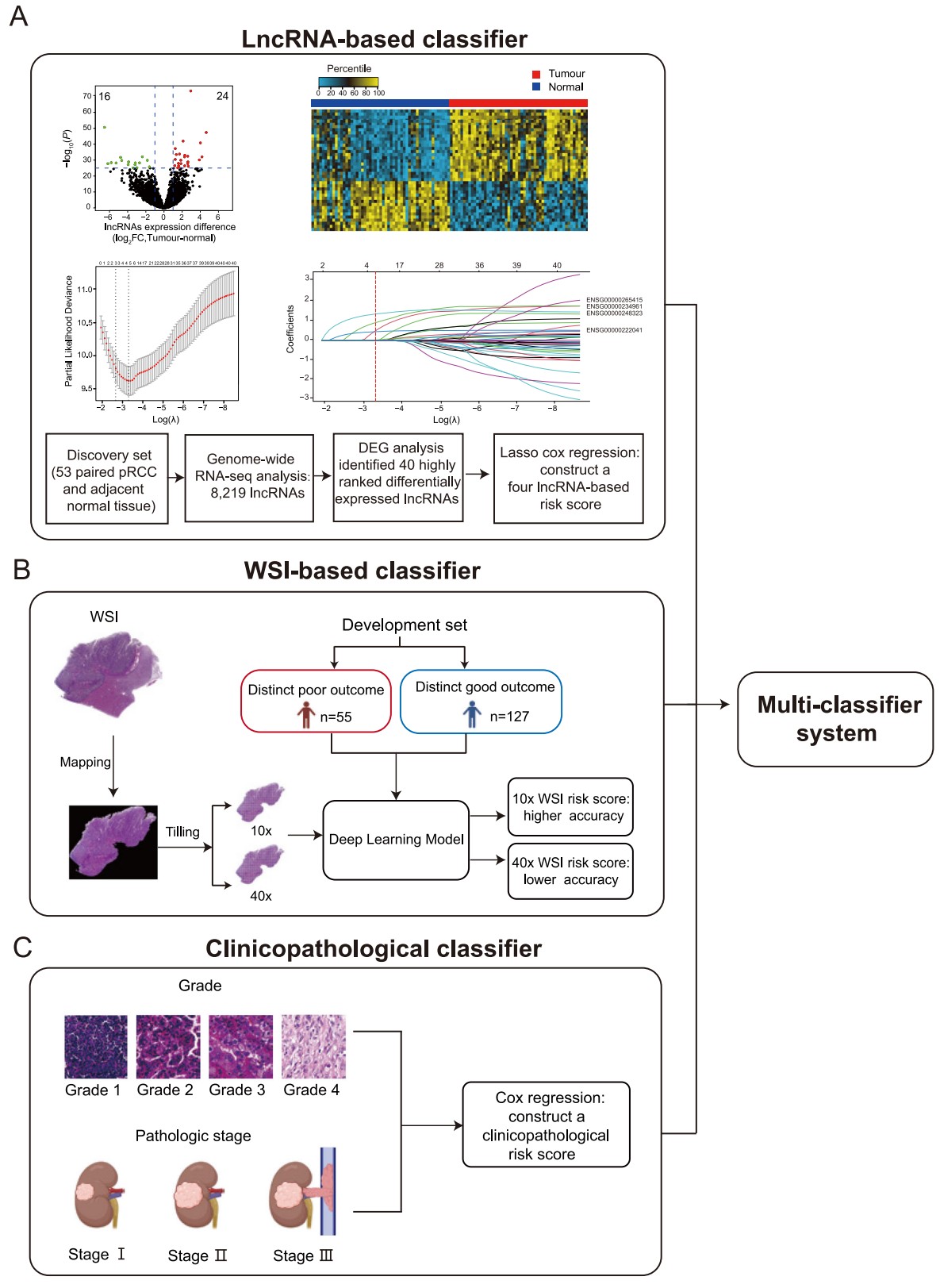

Grade 1, 2, 3, 4 were scored as 1, 2, 3, 4, respectively. Pathologic stage 1, 2, 3 were scored as 1, 2, 3, respectively (Fig. 1C; see Methods).

**Constructing a multi-classifier system in the training set**

Next, we developed a multi-classifier system integrating the four-lncRNA-based classifier, the WSI-based classifier, and the clinicopathological classifier using Cox regression coefficients:

$$
\begin{aligned}
\text{The multi} - \text{classifier risk score} =\;& 1.2924 \\
& \times \text{the lncRNA-based risk score} + 2.6315 \\
& \times \text{the WSI-based score} + 0.8646 \times (0.5670 \times \text{grade} \\
& + 0.5326 \times \text{stage}).
\end{aligned}
\tag{3}
$$

**Fig. 1 | Construction of the multi-classifier system.** lncRNA expression data, WSIs, and clinicopathological factors were used to develop the three classifiers respectively. We then integrated the three classifiers to develop a multi-classifier system. **A** The development of the lncRNA-based classifier. Upper left of panel: volcano plot comparing lncRNA expression in pRCC versus adjacent normal tissues ($n = 53$). Biological significance ($\log_2$ fold change (FC)) is depicted on the $x$ axis, and the statistical significance ($-\log_{10} P$) is depicted on the $y$ axis. Forty lncRNAs were identified with a $\log_2$ FC > 1, and the false discovery rate was <$10^{-25}$. Upper right of panel: heat map showing the expression level of 40 lncRNAs in 53 paired pRCCs. Middle left of panel: LASSO Cox regression analysis to select lncRNAs to include in the classifier. The two dotted vertical lines were drawn at the optimal values using the minimum criterion (right) and 1 minus the standard error (1–s.e.) criterion (left). Middle right of panel: LASSO coefficient profiles of the 40 differentially expressed lncRNAs. A vertical line was drawn at the optimal value using the minimum criterion, which resulted in four non-zero coefficients. Four lncRNAs were finally selected using the LASSO Cox regression model to build the four lncRNA-based score. Lower panel: flow chart. **B** The development of the WSI-based classifier using deep learning. **C** The development of the clinicopathological classifier. Pictures of pathologic stages were created with BioRender.com. Source data are provided as a Source Data file.

We found that, for the multi-classifier risk score, the C-index for predicting RFS was 0.831, which was significantly higher than that of any single classifier alone (C-index 0.661–0.760, $p < 0.01$ for all comparisons). Next, we divided patients in the training set into high-risk ($n = 191$) and low-risk ($n = 191$) groups, using the median multi-classifier risk score (4.1020) as the cutoff. Compared to patients in the low-risk group, patients in the high-risk group had shorter RFS (Hazard ratio (HR) 11.17, 95% CI 5.11–24.40, $p < 0.001$) (Fig. 2A) and shorter overall survival (OS) (HR 9.71, 95% CI 4.65–20.28, $p < 0.001$) (Supplementary Fig. 6).

**Validating the multi-classifier system in two independent sets**
To estimate the reproducibility and validity of this multi-classifier system, we tested it in two independent patient datasets: an independent validation set and the TCGA set (Table 1). The independent validation set included 207 cases from North China (two centers) and the TCGA set included 204 cases from the United States. The risk score for each patient in both independent sets was calculated using the same formula which are used in the training set (Fig. 2B, C). The multi-classifier risk score showed stable predictive accuracy that was similar in the independent validation set (C-index 0.831) and the TCGA set (C-index 0.858), and the accuracy was significantly higher both than that of any single classifier alone (C-index 0.642–0.777, $p < 0.05$ for all comparisons). Applying the same cutoff to establish high-risk and low-risk groups in the independent validation set, we found that patients in the high-risk group had shorter RFS (HR 12.85, 95% CI 4.61-35.84, $p < 0.001$) and OS (HR 10.90, 95% CI 3.90-30.46, $p < 0.001$) than patients in the low-risk group (Fig. 2B and Supplementary Fig. 6). Given the distinct lncRNA expression data type in the TCGA set, patients in the TCGA set were divided into high-risk and low-risk groups, using a distinct median risk score (5.3700) as the cutoff. Patients in the high-risk group in the TCGA set also had significantly shorter RFS (HR 8.54, 95% CI 3.02–24.14, $p < 0.001$) and OS (HR 9.21, 95% CI 2.16–39.32; $p = 0.003$) than patients in the low-risk group (Fig. 2C and Supplementary Fig. 6). After adjusting for clinical variables (age, sex, stage, and grade), the multi-classifier system remained an independent prognostic factor for predicting both RFS and OS in three sets (all $p < 0.05$, Supplementary Tables 4–7).

**Stratified analysis of the multi-classifier system and nomogram construction**
When stratified by clinical variables (age, sex, grade, and stage), the multi-classifier system was still a clinically and statistically significant prognostic model for predicting RFS and OS in all 793 patients with pRCC ($p < 0.05$ for all comparisons, Fig. 3 and Supplementary Figs. 7–9). High stage and grade were recognized as key determinants for selecting very-high-risk post-surgery patients who could benefit from adjuvant therapy[7]; however, our results showed that the RFS of patients with stage I and stage II pRCC in the multi-classifier risk-score-defined high-risk group was significantly shorter than that of patients with stage III pRCC in the low-risk group (HR 6.04 and 9.44, respectively, $p < 0.05$ for all comparisons) (Supplementary Fig. 10). Consistently, RFS was significantly worse for patients with grade 1/2 pRCC in the high-risk group compared to those with grade 3/4 pRCC in the low-risk group (HR 5.80, $p < 0.001$, Supplementary Fig. 10).

In addition, we constructed a nomogram that combined the lncRNA-based classifier, the WSI-based classifier, and the clinicopathological variables (stage and grade) to provide clinicians with a quantitative method to predict the 3-year, 5-year, and 7-year recurrence-free probabilities in a patient with pRCC (Fig. 4A). Calibration plots showed that the nomogram predict well in the training set, the independent validation set, and the TCGA set (Fig. 4B).

**Combining the multi-classifier risk score with other biomarkers in the TCGA dataset**
Several clusters of pRCC have been defined in the TCGA dataset based on CIMP, DNA methylation, mRNA, and microRNA profiling[6,11]. We compared these molecular cluster-based analyses with the multi-classifier risk score in the TCGA set, and we found that the multi-classifier risk score was significantly more accurate in predicting RFS compared to any of the molecular cluster-based approaches (C-index 0.858 for multi-classifier risk score compared to 0.569–0.660 for cluster-based analyses, $p < 0.001$ for all comparisons) (Fig. 5A). The TCGA Research Network reported that the CIMP hypermethylation pattern is a very important prognostic factor in pRCC[6,11]. Interestingly, in our study, we found that all the patients with tumors characterized by CIMP pattern had ultra-high multi-classifier-based risk scores (>12.0, Fig. 5B). The multi-classifier-based risk score of all the patients with tumors with CIMP pattern was >14.0, and it was <11.6 for all patients who did not display CIMP pattern. Strikingly, RFS was poorest among patients with pRCC with CIMP pattern and an ultra-high multi-classifier-based risk score (Fig. 5C). This suggests that our multi-classifier system is strongly correlated with CIMP and can accurately predict CIMP pattern.

## Discussion
In this study, we combined a four-lncRNA-based classifier, a WSI-based classifier, and a clinicopathological-based classifier to generate a multi-classifier risk score that is predictive of RFS. This multi-biomarker-based approach more accurately predicted RFS in patients with pRCC than any of the three classifiers alone, and it also outperformed the existing staging system. The multi-classifier-based risk score was developed and validated in three independent sets, which included patients from multiple centers in China and the United States, and the predictive accuracy of the approach was similar among patients from different regions of these countries. To our knowledge, this is the largest biomarker discovery project to date in pRCC.

The 2022 World Health Organization classification no longer recommends subclassification of pRCC into type 1 and type 2, and stage and grade are currently the two most relevant clinicopathological prognostic factors of pRCC. The prognosis of patients with high-stage and high-grade pRCC is usually worse than that of patients with low-stage and low-grade pRCC; tumor with high stage and/or high grade could benefit from adjuvant therapy[7]. However, our results showed that the prognosis of patients with pRCC stage I/II in the multi-classifier-defined high-risk group was significantly worse

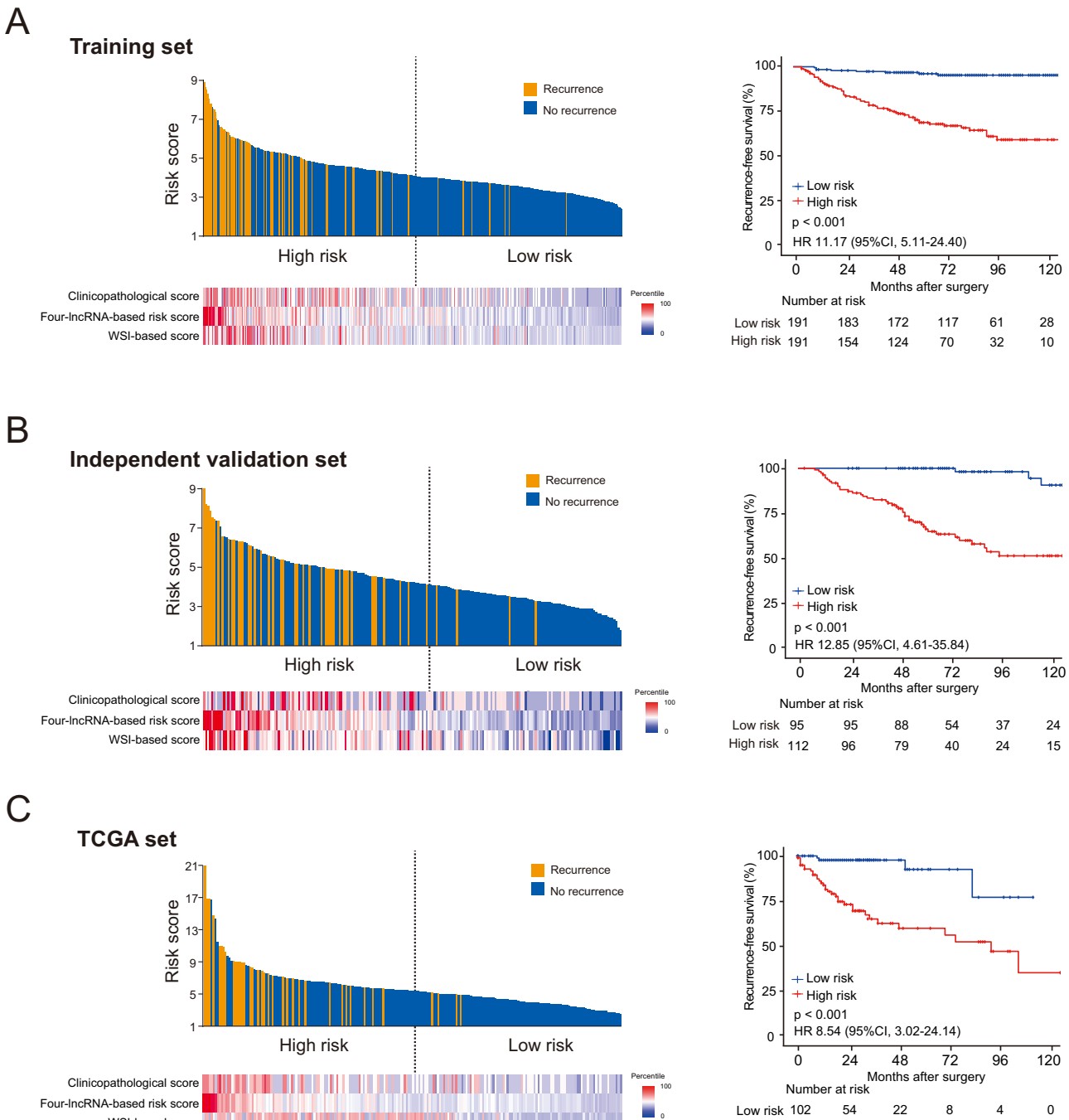

**Fig. 2 | Multi-classifier-based risk score and Kaplan–Meier survival analysis in the training set and two validation sets. A** The multi-classifier-based risk score and Kaplan–Meier survival analysis in the training set ($n = 382$). Upper left of panel: distribution of the multi-classifier-based risk scores and patient recurrence status. Lower left of panel: heat map showing the scores generated using each of the three classifiers independently. Right of panel: Kaplan–Meier survival analysis for RFS in patients with pRCC who were divided into low-risk and high-risk groups. **B**, **C** The multi-classifier-based risk scores and Kaplan–Meier survival analysis in the independent validation set ($n = 207$) and the TCGA set ($n = 204$), respectively. HRs, 95% CIs and two-sided $P$ values were calculated using the Cox proportional hazards model. Source data are provided as a Source Data file.

compared to patients with pRCC stage III in the multi-classifier-defined low-risk group. Moreover, the prognosis of patients with multi-classifier-defined high-risk grade 1/2 pRCC was worse than the prognosis for patients with multi-classifier-defined low-risk grade 3/4 pRCC. Thus, our multi-classifier-based risk score could be combined with current clinicopathological risk factors for pRCC to obtain more accurate prognostic data to inform personalized treatment strategy: identifying the low-risk subgroup of patients in the same clinical stage to prevent overtreatment where the absolute benefits of adjuvant

therapy are minimal relative to surgery alone; identifying the high-risk subgroup in the same clinical stage to prevent undertreatment where adjuvant therapy is needed to minimize the likelihood of recurrence.

A subtype of pRCC with CIMP hypermethylation is associated with particularly poor prognosis[6]. In the present study, we found that all of the patients with CIMP hypermethylation also had an ultra-high multi-classifier-defined risk score as well as very poor survival outcomes, indicating that these patients may benefit from more intensive treatment. Necrosis has been shown to be a prognostic factor in ccRCC.

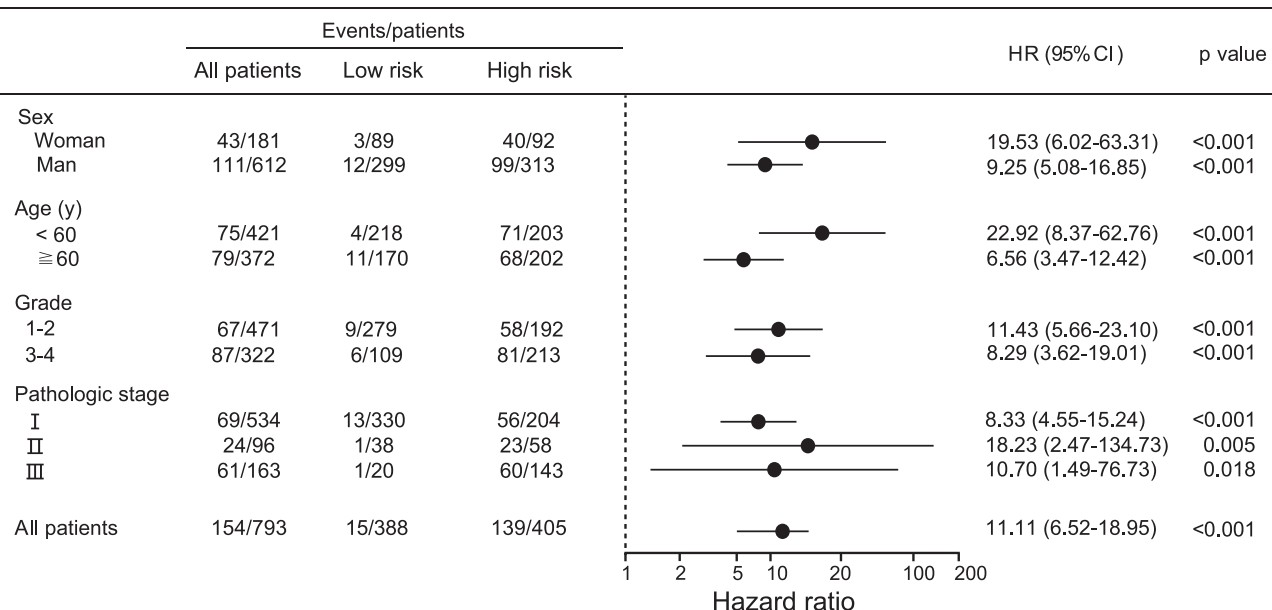

**Fig. 3 | HR of RFS for patients with pRCC predicted using the multi-classifier-based risk score in high-risk and low-risk groups.** The HR of RFS for all 793 patients with pRCC predicted using the multi-classifier-based risk score in subgroups stratified by clinical and pathological parameters. HRs, 95% CIs and two-sided *P* values were calculated using the Cox proportional hazards model. HRs are depicted as the central point for the error bars, while the 95% CI is represented by the length of the error bars. Source data are provided as a Source Data file.

However, there is currently no conclusive evidence that necrosis is prognostic in the pRCC context. In fact, in a study of 607 patients with pRCC, Leibovich et al.[5] reported that necrosis was not a prognostic factor, and a similar conclusion was reached by another study of 421 patients with pRCC from the Mayo Clinic[25]. Moreover, in the TCGA pRCC dataset used in this study, data regarding tumor necrosis are not available. For these reasons, we did not evaluate necrosis as a potential classifier in our study.

In recent years, increasing evidence has shown that lncRNA is involved in multi-level regulation of biological processes, and is considered a effective biomarker that can be stably examined in FFPE tissue. The feasibility of predicting cancer outcomes by detecting lncRNA expression by qRT–PCR assay from FFPE tissue samples was confirmed in many prognostic studies. Prensner et al.[16] used FFPE tissue samples from 1008 patients with localized prostate cancer and evaluated lncRNA expression profiles by microarray in the training cohort, which identified the lncRNA SChLAP1 as the highest-ranked overexpressed gene associated with cancer progression. Validation in three independent cohorts confirmed the prognostic value of SChLAP1. Ozawa et al.[26] assessed the relationship between the expression levels of 12 lncRNAs located in the 8q24.21 locus, which were detected using qRT–PCR analysis of FFPE tissue samples, and prognosis for patients with colorectal cancer. Two of these lncRNAs were identified and validated as reliable prognostic biomarkers for colorectal cancer. Qu et al.[27] developed an lncRNA-based signature of ccRCC that could be effectively identified through qRT–PCR analysis of FFPE tissue samples. In this study, our lncRNA-based classifier could predict patient survival and is applicable to routinely available FFPE tumor tissue from patients with pRCC. Moreover, the lncRNA expression profiles required for this classifier can be acquired not only through high-throughput sequencing but also via qRT–PCR assay, making our classifier practical and cost-effective to implement in clinical practice.

One of the distinctive characteristics of lncRNAs is their highly tissue-specific and cell-type-specific expression pattern, compared to mRNAs[28,29]. Qu et al.[27] compared a four-lncRNA-based signature with two established mRNA signatures (the 16-gene assay and ClearCode34) in three independent sets that included 1869 patients with ccRCC, and the four-lncRNA-based signature more accurately predicted OS than the two established mRNA signatures (C-index 0.74 vs. 0.64 and 0.66, respectively). Our lncRNA-based classifier also has higher predictive accuracy than the mRNA cluster in the TCGA set (C-index 0.777 vs. 0.608).

Of the four lncRNAs included in the lncRNA-based classifier, CYTOR (Ensembl ID: ENSG00000222041) is a well-studied lncRNA on chromosome 2, which acts as an oncogene in many cancers. It was shown to promote colon cancer metastasis in vitro and in vivo by interacting with β-catenin and drive colorectal cancer progression by interacting with NCL and Sam68[30,31]. Moreover, CYTOR is involved in chemotherapy resistance and epithelial–mesenchymal transition of oral squamous cell carcinoma[32]. LUCAT1 (Ensembl ID: ENSG00000248323) is located on chromosome 5 and it is noteworthy that this lncRNA is a significant prognostic factor for poor survival in ccRCC[27]. In colorectal cancer, LUCAT1 was determined to promote tumor proliferation by inhibiting the function of NCL and enhance chemotherapy resistance both in vitro and in vivo[33]. LUCAT1 was also reported to promote tumorigenesis in esophageal squamous cell carcinoma by regulating the stability of DNA methyltransferase 1[34]. AC099850.3 (Ensembl ID: ENSG00000265415) located on chromosome 17, acts as an oncogene in hepatocellular carcinoma and regulates tumor cell proliferation and invasion in vitro and in vivo through the PRR11/PI3K/AKT axis[35]. lnc-TRDMT1-5 (Ensembl ID: ENSG00000234961) is located on chromosome 10, and is positioned antisense to a well-known EMT marker, VIM. lnc-TRDMT1-5 was shown to correlate with poor survival in breast cancer[36].

Increasing studies have shown that deep learning combined with digital scanning of H&E-stained slides can identify histopathological or molecular features that are difficult to recognize with the human eye and could be used to develop diagnostic or prognostic cancer biomarkers[37–39]. A deep learning WSI-based model can predict survival in leiomyosarcoma and can help pathologists make more accurate diagnoses[24], and another WSI-based model precisely predicted the cancer-specific survival of patients with colorectal cancer[20]. In the present study, our WSI-based classifier discriminated patients with different risks of pRCC recurrence with high accuracy. This classifier is directly applicable to routine H&E-stained tissue sections, which

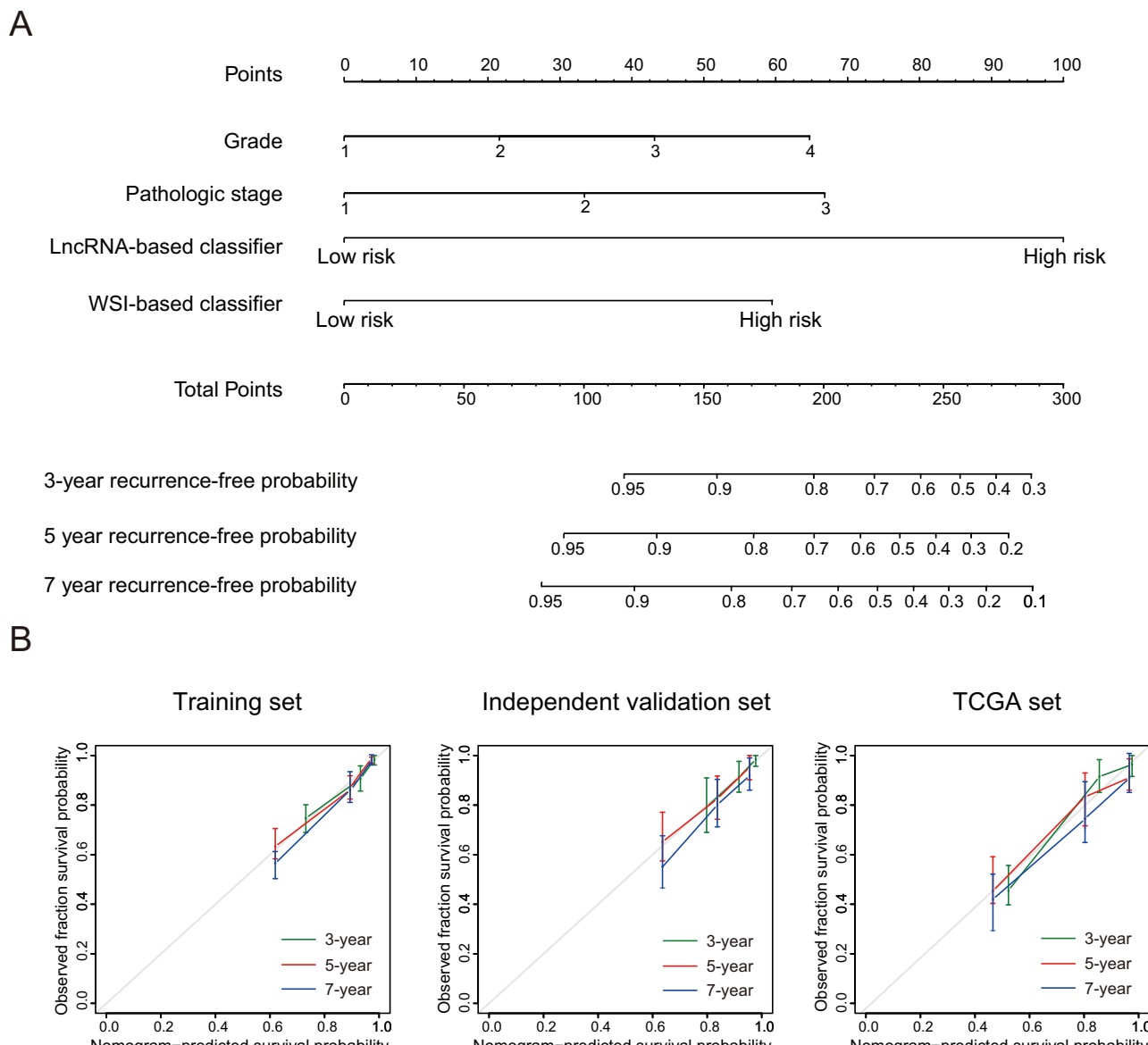

**Fig. 4 | The nomogram based on the lncRNA-based classifier, the WSI-based classifier, and the clinicopathological risk factors. A** The nomogram based on the lncRNA-based classifier, the WSI-based classifier, and clinicopathological risk factors for predicting the 3-year, 5-year, and 7-year recurrence-free probability for patients with pRCC after surgery. **B** Calibration curves of the nomogram to predict 3-year, 5-year, and 7-year RFS in the training set ($n = 382$), independent validation set ($n = 207$) and TCGA set ($n = 204$). The actual outcome is plotted on the $y$ axis, and the nomogram-predicted outcome is plotted on the $x$ axis. Model performance is shown relative to the 45° line, representing the performance of an ideal nomogram for which the predicted outcome perfectly corresponds with the actual outcome. The error bands represent the 95% CIs around the observed values. Source data are provided as a Source Data file.

makes it easy to translate for clinical application. Tumor heterogeneity is a critical factor that must be considered when investigating the molecular and histological characteristics of tumors. When developing our WSI-based classifier, we could only aim to minimize the impact of intra-tumor heterogeneity, as it is impossible to completely exclude it. In this study, we utilized a representative H&E-stained tumor slide (diagnostic slide), which chosen by pathologists for pathological diagnosis, represents the most characteristic view of the tumor. Given that slides at a 10× resolution provided better predictive accuracy than those at 40× resolution in our study, it suggests that incorporating a broader spectrum of histological information into deep learning image recognition could enhance model prediction accuracy. In future studies, we might increase the number of representative slides used, aiming to improve the predictive accuracy of our WSI-based classifier.

Several limitations of this study should be noted. First, the generalizability of our retrospective study is constrained because it only included patients from China and the United States. Although using the median risk score as a cutoff is a common practice in many studies, this approach's population-specific nature restricts its clinical application. Therefore, this multi-classifier system requires further validation through prospective studies in large-scale, multi-center clinical trials that encompass additional geographic regions before it can be widely applied in clinical settings. Second, we used a manual annotation method to delineate the tumor area in WSIs, which increases the workload of pathologists. In future studies, we will apply a convolutional neural network to automate the procedure for high-throughput clinical applications.

In summary, we developed and validated a practical multi-classifier system for patients with localized pRCC that can complement the current staging system to predict tumor recurrence with increased accuracy. Our multi-classifier-based risk score discriminates patients with localized pRCC according to risk of disease recurrence

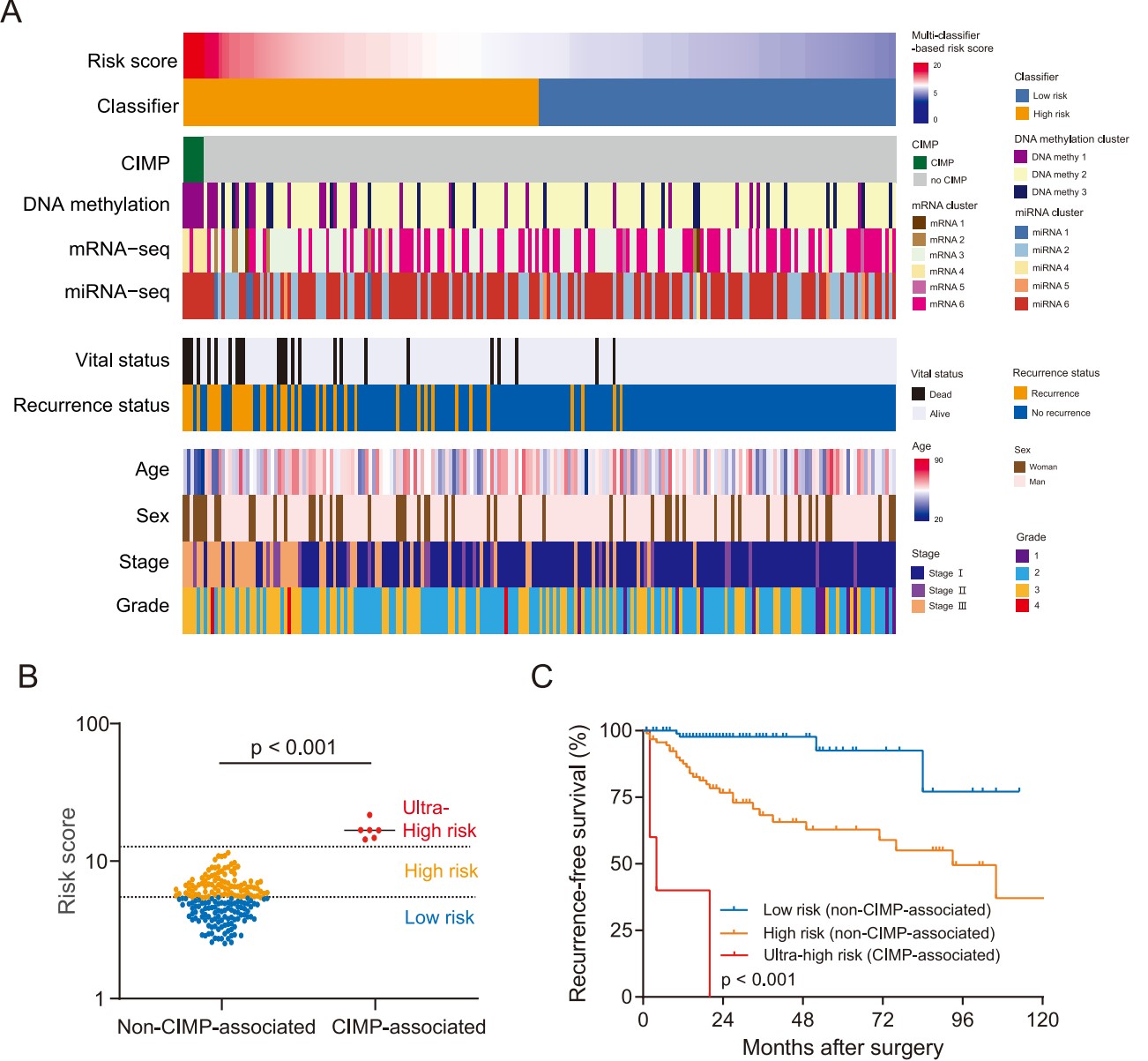

**Fig. 5 | Multi-classifier-based risk score predicts RFS in 204 patients with pRCC in the TCGA set. A** The heat map of 204 patients, which includes the multi-classifier-based risk score, established cluster-based molecular classifiers from TCGA, and clinical features. The cases are arranged according to the multi-classifier-based risk score. **B** The comparison between the multi-classifier-based risk score among patients according to whether their tumors had CIMP hypermethylation pattern using a scatter plot analyzed using two-sided unpaired Student's *t* test. The blue, orange, and red dots in the scatter plot represent low-risk, high-risk, and ultra-high-risk patients as determined using the multi-classifier-based risk score, respectively. **C** The Kaplan–Meier analysis of RFS according to whether the tumors had CIMP pattern. Patients with CIMP-associated tumors (*n* = 6) had significantly shorter RFS compared to patients with non-CIMP-associated tumors (*n* = 198), including patients in the low-risk group (*n* = 102) and high-risk group (*n* = 96) according to the multi-classifier system. *P* values were calculated with the log-rank test. Source data are provided as a Source Data file.

after surgery, thus enabling more informed decisions about adjuvant therapy.

## Methods
### Patients
This study was approved by the Institutional Review Boards of the First Affiliated Hospital of Sun Yat-sen University, Sun Yat-sen University Cancer Center, Renji Hospital of Shanghai Jiao Tong University, Peking University First Hospital, and Affiliated Yantai Yuhuangding Hospital of Qingdao University. The informed consent was waived because patients were not directly recruited for this study.

In this study, we used 589 formalin-fixed, paraffin-embedded (FFPE) tissue samples from 589 patients with pRCC who were treated between January 2008 and December 2018. Inclusion criteria were patients with sporadic unilateral pRCC, stage I–III, who underwent resection without neoadjuvant therapy or adjuvant therapy, and for whom clinicopathological characteristics, follow-up information, and fixed tumor tissue were available. Two genitourinary pathologists (Y.C. and B.L.) reassessed all of these samples. The training set (382 cases) included 189 patients from South China (First Affiliated Hospital and Cancer Center of Sun Yat-sen University) and 193 patients from East China (Renji Hospital of Shanghai Jiao Tong University). The independent validation set included 207 cases from North China and comprised 121 patients from Peking University First Hospital and 86 patients from Affiliated Yantai Yuhuangding Hospital of Qingdao University. For all patients in our cohorts from China, baseline imaging

assessments were conducted within 3 months following surgery. Beyond the initial assessments, patients diagnosed with stage I disease underwent annual evaluations until the occurrence of disease recurrence, metastasis, or death, whichever occurred first. For patients with stage II and stage III disease, evaluations were scheduled every 3–6 months during the first 3 years, followed by annual assessments until the occurrence of disease recurrence, metastasis, or death, whichever occurred first. Imaging assessments were collaboratively performed by urologists and radiologists at each site. CT scans (preferred) or MRIs (when CT was unavailable or impractical) were utilized for imaging the chest, abdomen, and pelvis. Additionally, bone scans, brain imaging, and other supplementary imaging procedures were undertaken as indicated by symptoms. For the TCGA set, RNA-seq data (level 3), clear diagnostic whole-slide images (WSIs) and clinical data were downloaded from the Genomic Data Commons portal (https://portal.gdc.cancer.gov/) on June 1, 2023. 204 cases with stage I–III, complete RNA-seq data, clear diagnostic WSIs and follow-up data were finally selected for further analysis. Cases with conflicting information were thoroughly re-evaluated and discussed again using all available information to reach a final diagnosis.

### Genome-wide RNA-seq data analysis

To generate lncRNA expression profiles, we obtained, as a discovery set, a panel of 53 fresh-frozen tumor samples with paired adjacent normal tissue from patients with pRCC treated between January 2016 and June 2020 at First Affiliated Hospital, Cancer Center of Sun Yat-sen University, and from Renji Hospital of Shanghai Jiao Tong University. Total RNA was extracted using TRIzol reagent (Invitrogen, Carlsbad, CA, USA). RNA purity and integrity were analyzed using a NanoDrop 2000 spectrophotometer (Thermo Fisher Scientific, Waltham, MA, USA) and an Agilent Bioanalyzer 2100 (Agilent Technologies, Santa Clara, CA, USA). The libraries were built and sequenced by CapitalBio Corporation (Beijing, China). Briefly, total RNA was subjected to removal of ribosomal RNA using a Ribo-Zero rRNA removal kit (Illumina, USA). A NEBNext Ultra II RNA Library Prep Kit for Illumina (New England Biolabs, Ipswich, MA, USA) was used to generate sequencing libraries following the manufacturer's protocols, and the library quality was monitored on an Agilent Bioanalyzer 2100 (Agilent). Fragments were finally sequenced on a HiSeq 2000 platform (Illumina).

Quality control and pre-processing of FASTQ files were done using fastp to obtain the clean reads (clean data)[40]. Obtained RNA-seq paired-end clean data were then analyzed using Hisat2[41], Samtools 1.9[42], Stringtie 1.3.5[43], and DESeq2[44]. Briefly, human reference genome (*Homo_sapiens.GRCh38.84.gtf.gz*) was obtained from Ensembl (ftp://ftp.ensembl.org/pub/release-84/gtf/homo_sapiens/Homo_sapiens.GRCh38.84.gtf.gz). Then, the reference genome index was created by the build-index function in the Hisat2 software (http://ccb.jhu.edu/software/hisat2) package with default options. The alignment of paired-end reads from each sample to the reference genome was performed using Hisat2 with default settings. After the alignment, the generated SAM files were sorted to BAM files using Samtools 1.9 (http://samtools.sourceforge.net). Subsequently, Stringtie 1.3.5 was used to assemble the transcripts using BAM files as inputs. We then used Stringtie and its prepDE.py to generate raw read count matrices for genes and transcripts. The genes were annotated by GENCODE version 22. Those lncRNAs located in the sex chromosomes were filtered. Only lncRNAs expressed in at least 80% of the tumor samples (counts ≥1) were included, and, finally, 8219 lncRNAs were selected for subsequent differential expression analysis. Differential expression analysis was performed using the DESeq2 package in R software. 40 lncRNAs were identified with a log2 fold change of more than 1, and the false discovery rate (FDR) was less than $10^{-25}$ (Supplementary Table 1). Raw sequencing data analyzed in this study have been uploaded to the Gene Expression Omnibus at the National Center of Biotechnology Information and can be found under accession number: GSE180777 or (https://www.ncbi.nlm.nih.gov/geo/query/acc.cgi?acc=GSE180777).

For the TCGA set, the expression of lncRNAs was derived from RNA-seq data downloaded from the Genomic Data Commons portal (https://portal.gdc.cancer.gov/).

### qRT–PCR

For each sample of FFPE tumor tissue from the training set and the independent validation set, serial sections were stained with hematoxylin and eosin for histological confirmation of the presence (>80%) of tumor cells. Three 20-μm tissue sections from each sample were used to obtain sufficient RNA. RNA was extracted with the Qiagen FFPE RNeasy kit and the HiPure FFPE RNA Kit (Magen, Guangzhou, China) following the manufacturer's instructions. For qRT–PCR, total RNA (2 μg) was reverse transcribed using PrimeScript RT Master Mix (Takara, Shiga, Japan), and qPCR was performed on triplicate samples in a SYBR Green Reaction Mix (Yeasen, Shanghai, China) with ABI 7900HT Fast Real Time PCR System (Applied Biosystems, Carlsbad, CA, USA). The sequences of the primers used in this study are listed in Supplementary Table S1, and the results were normalized to the expression levels of ACTB using the $2^{-\Delta Ct}$ method; ΔCt is the difference of Ct values between the lncRNA of target gene and the internal reference (ACTB).

### The construction of the lncRNA-based classifier

A multivariate LASSO Cox regression model was employed to select four lncRNAs, which were then used to generate a lncRNA-based risk score for RFS for each patient, as described in Eq. (1). The expression levels of the four lncRNAs for the training and independent validation sets were obtained from qRT-PCR assays, while for the TCGA set, they were extracted from RNA-seq data. Given the distinct data types, the applied cutoffs varied accordingly. In the training set, patients were divided into high-risk and low-risk groups using the median risk score of 0.9800 as the cutoff. This same cutoff was employed for classifying patients in the independent validation set into respective risk groups. For the TCGA set, patients were divided into high-risk and low-risk groups, using a distinct median risk score (1.8100) as the cutoff.

### Sample preparation and image pre-processing for WSI

A representative H&E-stained tumor slide (the one containing the highest grade of tumor in the specimen) was created for each patient in the training and independent validation sets[20,45]. Next, a WSI in the SVS file format was created for each representative slide using a KF-PRO-020 scanner (KFBIO, Ningbo, China), at 40 equivalent magnification (0.25 μm/pixel). Then, each representative slide was scanned with KF-PRO-020 scanner at 40× equivalent magnification to generate a WSI in the SVS file format. A WSI with the 40× resolution typically contained an order of 100,000 ×10,000 pixels—multiple orders of magnitude larger than images currently feasible for classification by deep learning methods[46,47]. The WSIs were annotated using the ASAP 1·8 platform, available at https://github.com/computationalpathologygroup/ASAP/releases. Next, a manual annotation method was applied to annotate the tissue regions and map the tumor area on all WSIs using polygons to draw the outline. This step was independently checked by another pathologist[48]. To preserve the prognostic information present at high resolution, WSIs were divided into multiple non-overlapping image regions known as tiles at 10× and 40× resolutions. Each pixel at 40× represents a physical size of approximately 0.24 × 0.24 μm. By creating a grid of potential tiles starting from the top-left corner of the slide picture, including areas outside the tumor segmentation, tiling was carried out. Candidates were tiles with an overall tissue percentage of more than 75% (compared to the whole area in each tile)[45]. Tiles were extracted with OpenSlide from level 0, converted to numpy arrays, and resized with OpenCV using the resize function (https://docs.opencv.org/3.4.0/da/d54/group_imgproc_transform.html), with interpolation

set to cv2.INTER_CUBIC for up-sampling and cv2.INTER_AREA for down-sampling, and saved in a lossless format (as TIFF files). To prepare each tile for the later prediction model, non-tissue-containing white background was removed from each tile using a series of RGB filters, and then color was normalized using the Macenko method[20].

## Construction and evaluation of the deep learning WSI-based classifier

182 cases from the training set with a so-called distinct outcome, either good or poor, were used as the development set to obtain clear ground truth for developing a prognostic score utilizing deep learning on digital pathological images. 76,348 tiles at 10× resolution and 1,789,634 tiles at 40× resolution in the developing cohort were used to train the deep learning models. Then, the developing cohort was randomly split into three sets—a discovery set (1,322,009 tiles from 136 WSIs), a tuning set (209,600 tiles from 18 WSIs), and a holdout internal test set (334,373 tiles from 28 WSIs)—for development of the outcome classifier[45]. The model was built around multiple instance learning and comprised a MobileNet V3 representation network[49,50], a Noisy-AND pooling function[51], and a fully connected classification network similar to the one used by Skrede et al.[20]. The entire network was trained end-to-end (i.e., directly from image to patient outcome), and each training iteration used a batch size of 32 collections with 64 tiles each. In brief, this convolutional neural network was a modified MobileNetV3 architecture that had been pre-trained on ImageNet and fine-tuned by transfer learning on the development set, which took tiles from an image and output a slide-level 4 probability (risk score) of tumor recurrence[20,45]. We added positional transforms, such as a horizontal flip and a rotation of 0°, 90°, 180°, or 270°, at random to the training data. We used a gradient approximation method that significantly lowers memory usage during training, which allowed us to use these tiles[20,45]. The Noisy-AND pooling function applied a trained non-linear function on tile representation averages. This function enhances robustness against tiles not representing the ground truth and, together with the large number of tiles, alleviates the issues of spatial heterogeneity. The network processed each tile in the WSI during inference. Using TensorFlow 2.4.0, the models were trained beyond apparent convergence, and each 10× model was evaluated at iteration 5000, 5500, and so on up to iteration 15,000. Each 40× model was evaluated at iterations 10,000, 11,000, and so on up to iteration 30,000. The models were selected from each network training using the performance in the tuning set from developing cohort with the C-index as metric, resulting in optimal models for each resolution. The predictive accuracy of the WSI-based score with the 10× resolution was higher than that with the 40× resolution.

We used the internal holdout test set from the development set for internal assessment, and we independently evaluated any potential performance variability related to randomization in data splitting using a four-fold cross-validation scheme. On the holdout test set, the WSI-based model with the 10× resolution was able to predict tumor recurrence possibilities with a C-index of 0.723 (95% CI 0.553–0.994), and, upon four-fold cross-validation in the development set, the C-index ranged from 0.634 to 0.721, with a mean of 0.688 across the four values. Furthermore, the WSI-based model with the 40× resolution had lower performance than that with the 10× resolution, which achieved a C-index of 0.675 (95% CI 0.514–0.727), and, upon four-fold cross validation in the development set, the C-index ranged from 0.581 to 0.643, with a mean of 0.609 across the four values. Thus, we chose the WSI-based model with the 10× resolution for the subsequent analyses. In the process of training and validating the multi-classifier system, 141,029 tiles at 10× resolution from 382 WSIs in the training set, 79,866 tiles at 10× resolution from 207 WSIs in the independent validation set, and 81,323 tiles at 10× resolution from 204 WSIs in the TCGA set were included for analysis. Using the WSI based model, we calculated a risk score (with 10× resolution) for each patient in the training, independent validation, and TCGA sets. Patients in all three sets were divided into high-risk and low-risk groups using the median risk score from the training set (0.2857) as the cutoff. The source code for the deep learning model is available online (https://github.com/guichengpeng1/WSI-based-deep-learning-classifier-in-papillary-renal-cell-carcinoma).

## Outcomes
The primary outcome was RFS, defined as the time from surgery to local recurrence and distant metastasis[10]. The secondary outcome was OS, defined as the time from surgery to death from any cause.

## Statistical analysis
LASSO Cox regression analysis was used to select the most useful prognostic markers among candidate lncRNA and to construct a lncRNA-based classifier for predicting the RFS of patients with pRCC in the training set. We used the Kaplan–Meier method to analyze the correlation between variables and survival. We used the Cox regression model for multivariate survival analysis and Cox regression coefficients to generate a prognostic model and a nomogram. HRs and 95% confidence intervals (CIs) were calculated using the Cox proportional hazards model. Harrell's C-indexes were calculated to examine discrimination[52]. All statistical tests were performed with R software version 4.1.0 (R Foundation for Statistical Computing, Vienna, Austria). Statistical significance was set at $p < 0.05$.

## Reporting summary
Further information on research design is available in the Nature Portfolio Reporting Summary linked to this article.

## Data availability
The Raw-sequencing data generated in this study have been deposited in the Gene Expression Omnibus database under accession code GSE180777. All whole-slide images and patient data from the TCGA cohort used in this study are available from the Genomic Data Commons Data Portal [https://portal.gdc.cancer.gov/]. Source data are provided with this paper. The WSIs and annotation data for the training set and independent validation set are subject to restrictions. These data were utilized with institutional permission via Institutional Review Board approval for the current study and are not publicly available due to patient privacy obligations. However, these data will be made available to interested research partners upon request to the lead contact (J-HL) or the Institute of Precision Medicine, First Affiliated Hospital, Sun Yat-sen University, Guangzhou, 510080, China. Access to the data requires a data transfer agreement, approved by the legal departments of the requesting researcher and by all legal departments of the institutions that provided data for the study, and an ethics clearance. Source data are provided with this paper.

## Code availability
The source code is available online (https://github.com/guichengpeng1/WSI-based-deep-learning-classifier-in-papillary-renal-cell-carcinoma)[53].

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

## Acknowledgements

This study was supported by grants from the Natural Science Foundation of China (award numbers: 81725016, 82373433 to J.-H.L., 82303308 to Y.-Z.X., 82072813 to W.-D.Z.) and the National Key Research and Development Program of China (award number: 2016YFC0902600 to J.-H.L.). We thank the TCGA for providing data.

## Author contributions

J.-H.L. designed the study. K.-B.H., Y.-Z.X., X.-S.L., H.-W.Z., Y.-H.P., H.H., F.-J.Z., R.-Y.L., Z.-H.F., W.C., Y.-F.Y., S.-W.X., G.-P.H., Q.T., K.O., J.-T.W., M.C., B.-J.D., Y.-R.H., J.Z., J.-P.G., F.-N.J. and W.-D.Z. obtained and assembled data. K.-B.H., C.-P.G., J.-H.W., J.-H.L., J.-Z.C., Y.-H.C., B.L., Y.C., X.-K.Z., W.-F.C., D.X., M.-Y.C., Z.-Y.J., G.-M.Q., C.-X.L., P.-X.L., Z.-P.L., J.-T.H. and W.X. analyzed and interpreted the data. K.-B.H., W.X., J.-H.W. and J.-H.L. wrote the manuscript, which was edited by all authors, who have approved the final version.

## Competing interests

The authors declare no competing interests.

## Additional information

¹Department of Urology, First Affiliated Hospital, Sun Yat-sen University, Guangzhou, China. ²Department of Urology, Sun Yat-sen University Cancer center, Guangzhou, China. ³State Key Laboratory of Oncology in South China, Guangdong Provincial Clinical Research Center for Cancer, Sun Yat-sen University Cancer center, Guangzhou, China. ⁴Department of Urology, Renji Hospital, School of Medicine, Shanghai Jiao Tong University, Shanghai, China. ⁵Department of Urology, Peking University First Hospital, Institute of Urology, Peking University, National Urological Cancer Center, Beijing, China. ⁶Department of Urology, Affiliated Yantai Yuhuangding Hospital, Qingdao University, Yantai, China. ⁷Department of Urology, Jiangmen Hospital, Sun Yat-sen University, Jiangmen, China. ⁸Department of Urology, The Third Affiliated Hospital of Soochow University, Changzhou, China. ⁹Department of Pathology, First Affiliated Hospital, Sun Yat-sen University, Guangzhou, China. ¹⁰Department of Pathology, Sun Yat-sen University Cancer center, Guangzhou, China. ¹¹Department of Urology, Guangdong Key Laboratory of Clinical Molecular Medicine and Diagnostics, Guangzhou First People's Hospital, School of Medicine, South China University of Technology, Guangzhou, China. ¹²Department of Pathology, Affiliated Yantai Yuhuangding Hospital, Qingdao University, Yantai, China. ¹³School of Mathematics and Computational Science, Sun Yat-sen University, Guangzhou, China. ¹⁴Institute of Precision Medicine, First Affiliated Hospital, Sun Yat-sen University, Guangzhou, China. ¹⁵Department of Internal Medicine and Department of Molecular Biology, University of Texas Southwestern Medical Center at Dallas, Dallas, TX, USA. ¹⁶Department of Urology, University of Texas Southwestern Medical Center at Dallas, Dallas, TX, USA. ¹⁷These authors contributed equally: Kang-Bo Huang, Cheng-Peng Gui, Yun-Ze Xu, Xue-Song Li, Hong-Wei Zhao, Jia-Zheng Cao. ✉e-mail: uroxuewei@163.com; weijh23@mail.sysu.edu.cn; luojunh@mail.sysu.edu.cn

