## [Peer Review File · Nature Communications]

A multi-classifier system integrated by clinico-histology-genomic analysis for predicting recurrence of papillary renal cell carcinomaREVIEWER COMMENTS

Reviewer #1 (Remarks to the Author): expertise in lncRNA signature development

The authors developed a multi-classifier system that integrated lncRNA features, histopathological WSI-based score, and clinicopathological score to predict the progression of pRCC. Although the authors showed the scoring system could be a predictor for progression of pRCC, the novelty and therapeutic significance of this system is limited.

1. What is the added novelty of this study compared to the previous study "Multimodal recurrence scoring system for prediction of clear cell renal cell carcinoma outcome: a discovery and validation study"?
2. What is the rationale to use lncRNA but not mRNA or other molecular features? Would the integration of selected mRNA/miRNA/methylation/genomic features with WSI score and clinicopathological result in compared or even better predictions?
3. The authors claim in abstract that the system can inform strategies for adjuvant therapy. Is there evidence to show the therapeutic significance of this system on adjuvant therapy?
4. The author should approach and describe the data with greater care, precision, and accuracy. For example, why did the author apply the same cutoff from training set for the validation set, but use a different median value for TCGA set (Line 185-190)? The p value for prediction of OS using the multi-classifier system is 0.004 or 0.042 instead of <0.001 in Supplementary Table 5 and 7 (Line 195). The number at risk and the plot for overall survival do not match (Fig. S6C). One of them must be wrong.

Reviewer #2 (Remarks to the Author): expertise in renal cancer -omics

This manuscript provides a well written analysis of a novel progression prediction algorithm for papillary renal cell carcinoma. It utilizes a multi-classifier system that combines clinical, histopathologic, and long non-coding RNA expression data and the accuracy of the predictions were confirmed using additional sample sets. Importantly, this prediction analysis concentrates on pRCC that, while rarer than clear cell RCC, makes up a significant percentage of RCC cases that are often underserved by analysis that concentrate on ccRCC.

I have a few comments concerning the manuscript listed below:

- 1) The cut-off for high risk and low risk is often defined in the analysis by simply being above or below the median risk value and this obviously differed between investigated cohorts. It would be very useful for the authors to also try to define an absolute risk value that could define these groups that could be universally used between cohorts for the same data type. This would undoubtedly produce some samples that are neither in the high or low groups, but this is not necessarily a problem. Analysis can be performed comparing either the defined high and low groups or the defined high group versus all other samples. This would also allow for a total evaluation of all samples across the cohorts. Defining a specific score above which a sample is considered high risk would be very useful for implementation of this prediction algorithm in actual samples.
- 2) In the WSI-based classifier a single slide was used to represent each tumor. This is reasonable for this initial analysis, but this seems likely to underrepresent features in heterogeneous tumors. It is known that ccRCC is a very heterogeneous tumor and it is possible this is true, to a lesser degree with pRCC. As this process is being performed in an automated manner it would be interesting to investigate multiple representative tissue sections from each tumor, especially the bigger tumors. The WSI-based score for each tumor could then be based on the highest scoring section within a tumor and it could be evaluated to see how this alters the accuracy of the calls. It is noticeably that the 10x sections were more accurate than the 40x sections and it is possible that this is due to more of the tumor being evaluated at 10x and more of the intratumoral heterogeneity was being perceived. Some comment on this would be very beneficial.
- 3) There are some existing clinicopathologic classifications of pRCC, did these show any similarity to the clinicopathological classifier defined in this project? This is slightly problematic as previous

classification included the type 1 and type 2 distinctions previously used in pRCC that are now not used in the WHO classification of pRCC.

4) In the discussion it is stated that the 4 selected lncRNAs have been previously shown to promote tumor progression or to correlate with cancer prognosis. More detail concerning these lncRNA and their known influence on cancer would be beneficial. Such as highlighting that LUCAT1 had been associated with the prognosis of localized clear cell RCC, while the others have been associated with other cancer types. The authors should also comment on the relative ease and practicality of evaluating a 4 lncRNA expression profile within tumors as this is specifically relevant to the usefulness of an assay like this.

5) Supplementary figure 2 does not seem to be referred to within the text of the results.

6) In lines 128-130, could the authors add the names of the lncRNAs (they are mentioned later) rather than the Ensembl identifiers.

7) In figure 1a, please add that the Lasso cox regression ended up producing a score based on 4 lncRNAs.

Reviewer #3 (Remarks to the Author): clinical expertise in papillary renal cell carcinoma

The authors should be applauded for focusing on papillary renal cell carcinoma (pRCC) – an understudied subtype that would benefit from more accurate prediction of recurrence probability for patients with stage I-III disease. However, there are notable limitations in this effort outlined below:

Major comments:

1. The authors used “progression-free survival” (PFS) as their main endpoint of interest in their study of patients with stage I-III disease. However, PFS is typically used in the metastatic setting (stage IV disease). Looking at their methods, PFS is defined by the authors as “the time from surgery to local recurrence, distant metastasis, or death from any cause”. However, that definition is typically used for recurrence-free survival (RFS). Sometimes we may use disease-free survival (DFS) which is a broader term that also encompasses the occurrence of a second cancer, which does not appear to have been part of the authors’ definition of their endpoint. Therefore, RFS rather than PFS appears to be the proper endpoint terminology the authors should use in their manuscript.

2. How was recurrence identified in the retrospective analyses of the Chinese cohorts? Was there a standardized imaging strategy or other surveillance strategy? The frequency of imaging can impact RFS and should be described in detail in the methods.

3. As noted by Frank Harrell (who invented the C-index) classification is not prediction (<https://www.fharrell.com/post/classification/>). Generalizable clinical prediction is a much harder task than classification. A recent paper in Science (Chekroud et al. Science 2024, PMID: 38207039) showcases how grave these challenges are. To their credit, the authors looked at the TCGA external database and showed calibration curves of the full nomogram which suggested that the nomogram overpredicts 5-year and 7-year survival probability when survival probability is > 0.6. It also appears that scenarios with lower survival probability of < 0.5 at 3 or 5 years (i.e., aggressive disease) were not included in the sample used for training and validation. In addition, the confidence bands for the prediction in the TCGA dataset are very wide. For example, the predicted seven year survival probability appears to range from less than 0.3 to up to 0.6. That is a substantial variation with clinically meaningful implications: decision for patients with survival probability of 30% at 7 years will be very different than for patients with survival probability of 60% at 7 years.

4. How does the final nomogram compare with other established nomograms such as the ASSURE

nomogram (Correa et al. Eur Urol. 2021, PMID: 33707112) available here:
[https://cancernomograms.com/nomograms/492 ?](https://cancernomograms.com/nomograms/492)

5. The 2022 WHO recommendations emphasize the use of molecular subtyping of PRCC based on oncogenic drivers (see Lobo et al. Histopathology 2022, PMID: 35596618). For example, certain fusion partners of TFE3 confer a worse prognosis than others in MiT family renal cell carcinoma (which was often previously diagnosed as PRCC in the past). Similarly, SMARCB1-deficient renal cell carcinoma and fumarate hydratase-deficient renal cell carcinoma have worse prognosis than the KRAS+ papillary renal neoplasm with reverse polarity, while all three of these entities would have been classified as PRCC in the past, including in the TCGA. What is the added value of the proposed nomogram once these molecular drivers have been accounted for?

RESPONSE TO REVIEWERS' COMMENTS

Reviewer #1:

1. *What is the added novelty of this study compared to the previous study*

“Multimodal recurrence scoring system for prediction of clear cell renal cell carcinoma outcome: a discovery and validation study”?

Our reply:

We are delighted that you took the time to read our previously published research (*Lancet Digit. Health, 2023*). Our current study is different from our previous study in the following ways:

(1) The molecular classifier in our previous study was developed using single-nucleotide polymorphism (SNP) data from The Cancer Genome Atlas (TCGA) database, without using our own high-throughput sequencing results. In contrast, the molecular classifier for papillary renal cell carcinoma (pRCC) in our current study was developed based on an analysis and selection from our own discovery set of 53 paired RNA sequencing (RNA-seq) results. This classifier underwent validation using the TCGA dataset, affirming its reliability.

To construct a reproducible and practical signature, the validation procession is as important as the developing procession. The validation process can proceed in one of two ways: one way is experimental detection and validation of the identified prognostic biomarkers; the other way is to assess and validate the identified prognostic biomarkers using an external cohort. The latter approach is more convincing and reliable than the former because it can prevent experimental bias on the part of the researcher. In our current study, we showed that the lncRNA-based classifier was reliable and reproducible because it was validated using both approaches with similar accuracy.

(2) Our previous study focused on clear cell renal cell carcinoma (ccRCC), which has distinct molecular and histological features compared to pRCC. For this reason, the TCGA database maintains separate repositories for ccRCC and pRCC (TCGA-KIRC and TCGA-KIRP, respectively). Thus, the prognostic models used for ccRCC cannot be directly applied to pRCC, making it essential to develop a prognostic model that is specifically designed for pRCC. Moreover, although molecular prognostic models for ccRCC are relatively abundant, similar models for pRCC remain scarce. Our current study represents the largest sample size and the most comprehensive effort to date in the field of biomarker research for pRCC.

2. *What is the rationale to use lncRNA but not mRNA or other molecular features? Would the integration of selected mRNA/miRNA/methylation/genomic features with WSI score and clinicopathological result in compared or even better predictions?*

Our reply:

The reviewer makes an important point that we are pleased to address. Prognostic signatures can be based on mRNA, they can also be based on epigenetic biomarkers, such as microRNA (miRNA), lncRNA and DNA methylation^{1, 2, 3}. We chose to analyze lncRNA signatures because of the following considerations:

- (1) **The exploration and validation of a prognostic signature-based lncRNA profile in pRCC is innovative.** Molecular characterization of pRCC was performed by the TCGA Research Network (*N. Engl. J. Med.*, 2016), which included copy number alterations, gene mutation, mRNA expression, DNA methylation patterns, and miRNA expression. From this, several prognostic signatures in pRCC were successfully constructed, including mRNA, DNA methylation, and miRNA signatures. However, in that study, lncRNA expression in pRCC was not analyzed in detail. Therefore, our study design based on the lncRNA profile complements the previous work of the TCGA Research Network⁴. In recent years, increasing evidence has shown that lncRNA has a relatively stable structure, is involved in multi-level regulation of biological processes, has potential in cancer targeted therapy, and is a novel biomarker that can effectively predict the clinical outcome of cancers, motivating us to pursue an lncRNA signature for pRCC.
- (2) **The prognostic accuracy of mRNA signature are not always better than or equal to that of epigenetic signatures.** One study compared a four-lncRNA-based signature with two notable mRNA signatures (the 16-gene assay and ClearCode34) in three independent sets that included 1,869 patients with ccRCC. The accuracy of predicting overall survival of the four-lncRNA-based signature is higher than two established mRNA signatures^{5, 6, 7}. In our current study, we compared our lncRNA-based signature with the mRNA signature and other molecular signatures that was developed by the TCGA Research Network in pRCC. Our lncRNA-based signature more accurately predicts clinical outcome than the mRNA signature, the miRNA signature, DNA methylation and the DNA copy number signature (C-index: 0.777 vs 0.569-0.660, $p < 0.001$ for all comparisons).
- (3) **Previous prognostic studies have demonstrated the relevance of epigenetic biomarkers in pRCC.** Indeed, the most important finding in the TCGA study (*N. Engl. J. Med.*, 2016) is not based on the mRNA profile but on the epigenetic profile; specifically, the study identifies a new molecular subtype of pRCC with CpG island methylator phenotype (CIMP)⁴. pRCC with the CIMP hypermethylation pattern has a particularly poor prognosis. Because both CpG methylation and lncRNA are epigenetic biomarkers, we further analyzed the

relationship between our multi-classifier system (including the lncRNA-based signature) and CIMP and found an interesting phenomenon. As shown in Figure 5B, C in the manuscript, all patients with the CIMP hypermethylation pattern fell within the multi-classifier-defined high-risk group, and pRCC patients with CIMP hypermethylation pattern in the multi-classifier-defined high-risk group had the poorest survival. Thus, our multi-classifier system can be combined with CIMP characterization in pRCC to more precisely predict clinical course.

References

1. Brock MV, *et al.* DNA methylation markers and early recurrence in stage I lung cancer. *N Engl J Med* **358**, 1118-1128 (2008).
2. Prensner JR, *et al.* Nomination and validation of the long noncoding RNA SCHLAP1 as a risk factor for metastatic prostate cancer progression: a multi-institutional high-throughput analysis. *Lancet Oncol* **15**, 1469-1480 (2014).
3. Ji J, *et al.* MicroRNA expression, survival, and response to interferon in liver cancer. *N Engl J Med* **361**, 1437-1447 (2009).
4. Linehan WM, *et al.* Comprehensive Molecular Characterization of Papillary Renal-Cell Carcinoma. *N Engl J Med* **374**, 135-145 (2016).
5. Qu L, *et al.* Prognostic Value of a Long Non-coding RNA Signature in Localized Clear Cell Renal Cell Carcinoma. *Eur Urol* **74**, 756-763 (2018).
6. Rini B, *et al.* A 16-gene assay to predict recurrence after surgery in localised renal cell carcinoma: development and validation studies. *Lancet Oncol* **16**, 676-685 (2015).
7. Brooks SA, *et al.* ClearCode34: A prognostic risk predictor for localized clear cell renal cell carcinoma. *Eur Urol* **66**, 77-84 (2014).

3. *The authors claim in abstract that the system can inform strategies for adjuvant therapy. Is there evidence to show the therapeutic significance of this system on adjuvant therapy?*

Our reply:

Thank you for raising this important point. Recent findings from the phase 3 EVEREST trial showed that patients with RCC (including pRCC) in the very-high-risk group could benefit from adjuvant therapy, whereas adjuvant treatment does not confer a survival benefit for patients in the intermediate-high-risk subgroup¹. This distinction suggests that patients categorized as very-high-risk could be candidates for adjuvant therapy, whereas patients categorized as intermediate-high-risk and low-risk patients might avoid such interventions, thus sparing them from the risks associated with overtreatment.

It is noteworthy that the risk stratification method used in the EVEREST study and in other RCC studies primarily relies on stage and grade². Our research takes this a step further not only by incorporating these parameters into our clinicopathological classifier but also by integrating into additional important classifiers: the lncRNA-based classifier and the whole-slide-image (WSI)-based classifier. The resulting multi-classifier system has markedly improved predictive accuracy beyond that of the clinicopathological classifier alone in the three sets (C-index 0.831-0.858 vs. 0.642-0.755, $p < 0.05$ for all comparisons).

To better guide the enrollment screening for clinical trials evaluating adjuvant therapy, we initiated a clinical trial (Multi-classifier System for Stratifying Stage III Papillary Renal Cell Carcinoma of Receiving Adjuvant Therapy, NCT06146777) that employs our multi-classifier system for selecting patients with pRCC for adjuvant treatment.

References

1. Ryan CW, *et al.* Adjuvant everolimus after surgery for renal cell carcinoma (EVEREST): a double-blind, placebo-controlled, randomised, phase 3 trial. *Lancet* **402**, 1043-1051 (2023).
2. Choueiri TK, *et al.* Adjuvant Pembrolizumab after Nephrectomy in Renal-Cell Carcinoma. *N Engl J Med* **385**, 683-694 (2021).

4. *The author should approach and describe the data with greater care, precision, and accuracy. For example, why did the author apply the same cutoff from training set for the validation set, but use a different median value for TCGA set (Line 185-190)? The p value for prediction of OS using the multi-classifier system is 0.004 or 0.042 instead of <0.001 in Supplementary Table 5 and 7 (Line 195). The number at risk and the plot for overall survival do not match (Fig. S6C). One of them must be wrong.*

Our reply:

Thank you for this careful review and these suggestions.

(1) The reason for applying different cutoff values between TCGA set and the other

two sets is due to the variation in the sources of the lncRNA expression data. The expression levels of lncRNAs in both the training set and the independent validation set were derived from qRT-PCR results, whereas those in the TCGA set were obtained from RNA-seq data (Supplementary Fig. 2). This resulted in variations in the measurement units used for the expression levels of the four lncRNAs between the TCGA set and the other two sets. Thus, we applied the same cutoff to the training set and the independent validation set, but a different cutoff was necessary for the TCGA set. The same cutoff value for the WSI-based classifier was applied to all three sets, given that their data sources were exclusively WSIs.

In the initial manuscript, we placed the cutoff values for the lncRNA-based classifier and the WSI-based classifier in the legends of Supplementary Fig. 3 and 5, respectively, while the cutoff values for the multi-classifier system were located in the Results section (Line 166-168 and 183-189). To better clarify why we set these cutoff values and to ensure that these values are easier to locate, we added corresponding descriptions in the Results section and the Methods section of the revised manuscript:

Cutoff values for the multi-classifier system:

Training set: Next, we divided patients in the training set into high-risk (n=191) and low-risk (n=191) groups, using the median multi-classifier risk score (4.1020) as the cutoff (Line 166-168). Independent validation set and the TCGA set: Applying the same cutoff to establish high-risk and low-risk groups in the independent validation set, we found that patients in the high-risk group had shorter RFS (HR 12.85, 95% CI 4.61-35.84, $p < 0.001$) and OS (HR 10.90, 95% CI 3.90-30.46, $p < 0.001$) than patients in the low-risk group (Fig. 2B and Supplementary Fig. 6). Given the distinct lncRNA expression data type in the TCGA set, patients in the TCGA set were divided into high-risk and low-risk groups, using a distinct median risk score (5.3700) as the cutoff (Line 183-189).

Cutoff values for the lncRNA-based classifier: Given the distinct data types, the applied cutoffs varied accordingly. In the training set, patients were divided into high-risk and low-risk groups using the median risk score of 0.9800 as the cutoff. This same cutoff was employed for classifying patients in the independent validation set into respective risk groups. For the TCGA set, patients were divided into high-risk and low-risk groups, using a distinct median risk score (1.8100) as the cutoff (Line 639-644).

Cutoff values for the WSI-based classifier: Patients in all three sets were divided into high-risk and low-risk groups using the median risk score from the training set (0.2857) as the cutoff (Line 723-725).

- (2) The p-value for the multi-classifier system in the TCGA set presented in Supplementary Fig. 6 was originally calculated using the log-rank test in Kaplan-Meier analysis, whereas the p-values in Supplementary Tables 5 and 7 were calculated using Cox regression analysis. To avoid any misunderstanding due to

the use of different statistical methods, in the revised manuscript, we also calculated the p-values in Supplementary Fig. 6 using Cox regression analysis.

Supplementary Fig. 6 Kaplan-Meier survival analysis for OS of the multi-classifier system in the three sets.

Reviewer #2:

1. The cut-off for high risk and low risk is often defined in the analysis by simply being above or below the median risk value and this obvious differed between investigated cohorts. It would be very useful for the authors to also try to define an absolute risk value that could define these groups that could be universally used between cohorts for the same data type. This would undoubtedly produce some samples that are neither in the high or low groups, but this is not necessarily a problem. Analysis can be performed comparing either the defined high and low groups or the defined high group verses all other samples. This would also allow for a total evaluation of all samples across the cohorts. Defining a specific score above which a sample is considered high risk would be very useful for implementation of this prediction algorithm in actual samples.

Our reply:

Thank you for this valuable suggestion. The reason for applying different cutoff values between the TCGA set and the other two sets is due to the variation in the sources of the lncRNA expression data: the expression levels of lncRNAs in both the training set and the independent validation set were derived from qRT-PCR results, whereas those in the TCGA set were obtained from RNA-seq data.

Initially, we had placed the cutoff values for the lncRNA-based classifier and the WSI-based classifier in the legends of Supplementary Figs. 3 and 5, respectively, while the cutoff values for the multi-classifier system were located in the Results section (Line 166-168 and 183-189). To improve accessibility and to ensure that these values are easier to locate, we incorporated corresponding descriptions in the Results section and the Methods section of the revised manuscript:

Cutoff values for the multi-classifier system:

Training set: Next, we divided patients in the training set into high-risk (n=191) and low-risk (n=191) groups, using the median multi-classifier risk score

(4.1020) as the cutoff (Line 166-168). Independent validation set and the TCGA set: Applying the same cutoff to establish high-risk and low-risk groups in the independent validation set, we found that patients in the high-risk group had shorter RFS (HR 12.85, 95% CI 4.61-35.84, $p < 0.001$) and OS (HR 10.90, 95% CI 3.90-30.46, $p < 0.001$) than patients in the low-risk group (Fig. 2B and Supplementary Fig. 6). Given the distinct lncRNA expression data type in the TCGA set, patients in the TCGA set were divided into high-risk and low-risk groups, using a distinct median risk score (5.3700) as the cutoff (Line 183-189).

Cutoff values for the lncRNA-based classifier: Given the distinct data types, the applied cutoffs varied accordingly. In the training set, patients were divided into high-risk and low-risk groups using the median risk score of 0.9800 as the cutoff. This same cutoff was employed for classifying patients in the independent validation set into respective risk groups. For the TCGA set, patients were divided into high-risk and low-risk groups, using a distinct median risk score (1.8100) as the cutoff (Line 639-644).

Cutoff values for the WSI-based classifier: Patients in all three sets were divided into high-risk and low-risk groups using the median risk score from the training set (0.2857) as the cutoff (Line 723-725).

By applying these cutoff values, we were able to categorize patients into high-risk and low-risk groups across all three cohorts. This allowed for a comprehensive evaluation of all patients in these groups, including forest plots (Fig. 3 and Supplementary Fig. 8) and other stratified analyses (Supplementary Figs. 7 and 9).

2. In the WSI-based classifier a single slide was used to represent each tumor. This is reasonable for this initial analysis, but this seems likely to underrepresent features in heterogenous tumors. It is known that ccRCC is a very heterogenous tumors and it is possible this is true, to a lesser degree with pRCC. As this process is being performed in an automated manner it would be interesting to investigate multiple representative tissue sections from each tumor, especially the bigger tumors. The WSI-based score for each tumor could then be based on the highest scoring section within a tumor and it could be evaluated to see how this alters the accuracy of the calls. It is noticeably that the 10x sections were more accurate than the 40x sections and it is possible that this is due to more of the tumor being evaluated at 10x and more of the intratumoral heterogeneity was being perceived. Some comment on this would be very beneficial.

Our reply:

This is an excellent question that we are pleased to address. An increasing number of studies are using H&E-stained slides for WSI-based AI prediction research on large solid tumors^{1, 2, 3}. Typically, the H&E-stained slides selected for these studies are representative slides. Within large solid tumors, there are regions with varying degrees of malignancy—some with relatively lower malignancy and others with higher malignancy. Pathologists select the representative slides by comparing slides from different areas of the tumor, choosing those that include the most invasive tumor regions. This principle guides the selection of representative slides in our research as

well as in the TCGA database, to ensure that the representative slides reflect the most malignant aspects of the solid tumors.

Reference

1. Yamashita R, *et al.* Deep learning model for the prediction of microsatellite instability in colorectal cancer: a diagnostic study. *Lancet Oncol* **22**, 132-141 (2021).
2. Zheng X, *et al.* A deep learning model and human-machine fusion for prediction of EBV-associated gastric cancer from histopathology. *Nat Commun* **13**, 2790 (2022).
3. Foersch S, *et al.* Deep learning for diagnosis and survival prediction in soft tissue sarcoma. *Ann Oncol* **32**, 1178-1187 (2021).

3. There are some existing clinicopathologic classifications of pRCC, did these show any similarity to the clinicopathological classifier defined in this project? This is slightly problematic as previous classification included the type 1 and type 2 distinctions previously used in pRCC that are now not used in the WHO classification of pRCC.

Our reply:

Thank you for this insightful comment. As the World Health Organization classification no longer recommends subclassification of pRCC into type 1 and type 2, incorporating histologic subtype into a clinicopathological classifier no longer aligns with the current consensus. Several existing clinicopathological models of pRCC use grade as a risk factor instead of histologic subtype. For instance, Leibovich et al.¹ identified grade, perinephric or renal sinus fat invasion, and tumor thrombus as the three main prognostic factors for post-surgery pRCC. The VENUSS score² includes tumor size, T stage, N stage, grade, and venous tumor thrombus as risk factors. Besides grade, the other risk factors mentioned in these models, such as tumor size, T stage, N stage, tumor thrombus, and renal sinus fat invasion, are components of the TNM staging system for RCC, hence falling under the category of pathologic stage.

The risk factors contained within our clinicopathological classifier are grade and pathologic stage, thus sharing similarities with previous clinicopathological models, and ensuring that our approach encompasses the currently identified key clinical and pathologic factors. Moreover, we applied the Leibovich model and the VENUSS model in our two cohorts (the TCGA set has no information for perinephric or renal sinus fat invasion or tumor thrombus), and we found that our clinicopathological classifier exhibits similar predictive accuracy compared to these two models (The following table).

Table Comparison between our clinicopathological classifier and other clinicopathological models in the Training and independent validation sets in our study

	Leibovich model	VENUSS model	Our clinicopathological classifier
C-index for predicting RFS	0.638 (p=0.429)	0.664 (p=0.232)	0.643
C-index for predicting OS	0.624 (p=0.311)	0.657 (p=0.282)	0.639

*The p-value is calculated to evaluate the statistical significance between this C-index and the C-index of our clinicopathological classifier.

Reference

1. Leibovich BC, et al. Predicting Oncologic Outcomes in Renal Cell Carcinoma After Surgery. *Eur Urol* **73**, 772-780 (2018).
2. Klatte T, et al. The VENUSS prognostic model to predict disease recurrence following surgery for non-metastatic papillary renal cell carcinoma: development and evaluation using the ASSURE prospective clinical trial cohort. *BMC Med* **17**, 182 (2019).

4. In the discussion it is stated that the 4 selected lncRNAs have been previously shown to promote tumor progression or to correlate with cancer prognosis. More detail concerning these lncRNA and their known influence on cancer would be beneficial. Such as highlighting that LUCAT1 had been associated with the prognosis of localized clear cell RCC, while the others have been associated with other cancer types. The authors should also comment on the relative ease and practicality of evaluating a 4 lncRNA expression profile within tumors as this is specifically relevant to the usefulness of an assay like this.

Our reply:

Thank you for this valuable suggestion.

- (1) In the revised manuscript, we added a detailed discussion concerning the four lncRNAs (Line 300-317): Of the four lncRNAs included in the lncRNA-based classifier, CYTOR (Ensembl ID: ENSG00000222041) is a well-studied lncRNA on chromosome 2, which acts as an oncogene in many cancers. It was shown to promote colon cancer metastasis in vitro and in vivo by interacting with β -catenin and drive colorectal cancer progression by interacting with NCL and Sam68.^{31,32} Moreover, CYTOR is involved in chemotherapy resistance and epithelial–mesenchymal transition of oral squamous cell carcinoma.³³ LUCAT1 (Ensembl ID: ENSG00000248323) is located on chromosome 5 and it is noteworthy that this lncRNA is a significant prognostic factor for poor survival in ccRCC.²⁸ In colorectal cancer, LUCAT1 was determined to promote tumor proliferation by inhibiting the function of NCL and enhance chemotherapy resistance both in vitro and in vivo.³⁴ LUCAT1 was also reported to promote tumorigenesis in esophageal squamous cell carcinoma by regulating the stability of DNA methyltransferase 1.³⁵ AC099850.3 (Ensembl ID: ENSG00000265415) located on chromosome 17, acts as an oncogene in hepatocellular carcinoma and regulates tumor cell proliferation and invasion in vitro and in vivo through the PRR11/PI3K/AKT axis.³⁶ lnc-TRDMT1-5 (Ensembl ID: ENSG00000234961) is located on chromosome 10, and is positioned antisense to a well-known EMT marker, VIM. lnc-TRDMT1-5 was shown to correlate with poor survival in breast cancer.³⁷
- (2) We also added another paragraph to discuss the relative ease and practicality of using the lncRNA-based classifier (Line 272-291): In recent years, increasing evidence has shown that lncRNA is involved in multi-level regulation of biological processes, and is considered a novel effective biomarker that can be stably examined in FFPE tissue. The feasibility of predicting cancer outcomes by detecting lncRNA expression by qRT–PCR assay from FFPE tissue samples was confirmed in many prognostic studies. Prensner et al.¹ used FFPE tissue samples from 1,008 patients with localized prostate cancer and evaluated lncRNA expression profiles by microarray in the training cohort, which identified the lncRNA SChLAP1 as the highest-ranked overexpressed gene associated with cancer progression. Validation in three independent cohorts confirmed the prognostic value of SChLAP1. Ozawa et al.² assessed the relationship between the

expression levels of 12 lncRNAs located in the 8q24.21 locus, which were detected using qRT-PCR analysis of FFPE tissue samples, and prognosis for patients with colorectal cancer. Two of these lncRNAs were identified and validated as reliable prognostic biomarkers for colorectal cancer. Qu et al.³ developed an lncRNA-based signature of ccRCC that could be effectively identified through qRT-PCR analysis of FFPE tissue samples. In this study, our lncRNA-based classifier could predict patient survival and is applicable to routinely available FFPE tumor tissue from patients with pRCC. Moreover, the lncRNA expression profiles required for this classifier can be acquired not only through high-throughput sequencing but also via qRT-PCR assay, making our classifier practical and cost-effective to implement in clinical practice.

References

1. Prensner JR, *et al.* Nomination and validation of the long noncoding RNA SCHLAP1 as a risk factor for metastatic prostate cancer progression: a multi-institutional high-throughput analysis. *Lancet Oncol* **15**, 1469-1480 (2014).
2. Ozawa T, *et al.* CCAT1 and CCAT2 long noncoding RNAs, located within the 8q.24.21 'gene desert', serve as important prognostic biomarkers in colorectal cancer. *Ann Oncol* **28**, 1882-1888 (2017).
3. Qu L, *et al.* Prognostic Value of a Long Non-coding RNA Signature in Localized Clear Cell Renal Cell Carcinoma. *Eur Urol* **74**, 756-763 (2018).

5. *Supplementary figure 2 does not seem to be referred to within the text of the results.*

Our reply:

Thank you for pointing this out. We refer to Supplementary Fig. 2 in the revised manuscript: The expression of the four lncRNAs is shown in Supplementary Fig. 2 (Line 128).

6. *In lines 128-130, could the authors add the names of the lncRNAs (they are mentioned later) rather than the Ensembl identifiers.*

Our reply:

Thank you for this suggestion. We replaced the lncRNA Ensembl IDs with their corresponding gene names in the revised manuscript (Line 126-127): Four-lncRNA-based risk score = (0.4537 × AC099850.3) + (0.8549 × lnc-TRDMT1-5) + (0.4143 × CYTOR) + (1.2739 × LUCAT1).

7. In figure 1a, please add that the Lassa cox regression ended up producing a score based on 4 lncRNAs.

Our reply:

Thank you for your suggestion. We incorporated this information into Fig. 1A and into the figure legend of this figure (Line 369-370) in the revised version of our manuscript.

Figure 1A The development of the lncRNA-based classifier.

Reviewer #3:

1. The authors used “progression-free survival” (PFS) as their main endpoint of interest in their study of patients with stage I-III disease. However, PFS is typically used in the metastatic setting (stage IV disease). Looking at their methods, PFS is defined by the authors as “the time from surgery to local recurrence, distant metastasis, or death from any cause”. However, that definition is typically used for recurrence-free survival (RFS). Sometimes we may use disease-free survival (DFS) which is a broader term that also encompasses the occurrence of a second cancer, which does not appear to have been part of the authors’ definition of their endpoint. Therefore, RFS rather than PFS appears to be the proper endpoint terminology the authors should use in their manuscript.

Our reply:

We appreciate this valuable suggestion. PFS, RFS, and DFS are all endpoints used in prognostic models in RCC research^{1, 2, 3}. Due to the lack of information on second cancer occurrences in the TCGA dataset, DFS is not available in the TCGA set. Initially, we opted for PFS as our research endpoint because we referenced the prior largest sample size study (607 cases) used for a pRCC prognostic model (the Leibovich model), which predicts PFS of patients with pRCC after surgery. Following the reviewer's suggestion, we attempted to use RFS as the primary endpoint to rebuild the model and reassess its predictive accuracy. We found that the predictive accuracy of the new nomogram improved within the TCGA set. Accordingly, we revised our study's primary endpoint to RFS. Corresponding modifications were made throughout the relevant sections of the revised manuscript.

References

1. Leibovich BC, *et al.* Predicting Oncologic Outcomes in Renal Cell Carcinoma After Surgery. *Eur Urol* **73**, 772-780 (2018).
2. Brooks SA, *et al.* ClearCode34: A prognostic risk predictor for localized clear cell renal cell carcinoma. *Eur Urol* **66**, 77-84 (2014).
3. Correa AF, *et al.* Predicting Disease Recurrence, Early Progression, and Overall Survival Following Surgical Resection for High-risk Localized and Locally Advanced Renal Cell Carcinoma. *Eur Urol* **80**, 20-31 (2021).

2. How was recurrence identified in the retrospective analyses of the Chinese cohorts? Was there a standardized imaging strategy or other surveillance strategy? The frequency of imaging can impact RFS and should be described in detail in the methods.

Our reply:

In response to the comments regarding postoperative surveillance, we detailed our strategy for regular imaging in the revised manuscript (Line 556-566). Specifically, our follow-up protocol for the cohorts in China is as follows:

For all patients in our cohorts from China, baseline imaging assessments were conducted within 3 months after surgery. Beyond the initial assessments, patients diagnosed with stage I disease underwent annual evaluations until disease recurrence, metastasis, or death, whichever occurred first. For patients with stage II and stage III disease, evaluations were scheduled every 3-6 months during the first 3 years, followed by annual assessments until disease recurrence, metastasis, or death, whichever occurred first.

Imaging assessments were collaboratively performed by urologists and radiologists at each site. CT scans (preferred) or MRIs (when CT was unavailable or impractical) were used for imaging the chest, abdomen, and pelvis. Additionally, bone

scans, brain imaging, and other supplementary imaging procedures were undertaken as indicated by symptoms.

3. As noted by Frank Harrell (who invented the C-index) classification is not prediction (<https://www.fharrell.com/post/classification/>). Generalizable clinical prediction is a much harder task than classification. A recent paper in Science (Chekroud et al. Science 2024, PMID: 38207039) showcases how grave these challenges are. To their credit, the authors looked at the TCGA external database and showed calibration curves of the full nomogram which suggested that the nomogram overpredicts 5-year and 7-year survival probability when survival probability is > 0.6 . It also appears that scenarios with lower survival probability of < 0.5 at 3 or 5 years (i.e., aggressive disease) were not included in the sample used for training and validation. In addition, the confidence bands for the prediction in the TCGA dataset are very wide. For example, the predicted seven year survival probability appears to range from less than 0.3 to up to 0.6. That is a substantial variation with clinically meaningful implications: decision for patients with survival probability of 30% at 7 years will be very different than for patients with survival probability of 60% at 7 years.

Our reply:

Thank you for highlighting this important point. The TCGA set (n=204) encompasses different ethnicities and populations, and presents a comparatively smaller number of cases than the training set (n=382). This discrepancy could potentially result in overprediction or underprediction for certain patients and contribute to relatively broad confidence intervals. After revising the primary endpoint of our study to RFS, we observed an improvement in the predictive accuracy of our nomogram in the TCGA set. As illustrated in Fig. 4B of the revised manuscript, using RFS as the primary endpoint significantly reduced overprediction of the nomogram, particularly when survival probability is greater than 0.6, and the width of confidence bands was also notably decreased.

Although our study is the largest biomarker discovery project to date in pRCC, the absolute numbers in our cohorts remain somewhat limited. We aim to increase the sample size (especially in Western population) in future work to increase further enhancing the predictive accuracy of our nomogram.

Fig. 4B Calibration curves of the nomogram to predict 3-year, 5-year, and 7-year RFS. The actual outcome is plotted on the y axis, and the nomogram-predicted outcome is plotted on the x axis. Model performance is shown relative to the 45° line, representing the performance of an ideal nomogram for which the predicted outcome perfectly corresponds with the actual outcome.

4. How does the final nomogram compare with other established nomograms such as the ASSURE nomogram (Correa et al. Eur Urol. 2021, PMID: 33707112) available here: <https://cancernomograms.com/nomograms/492> ?

Our reply:

We appreciate this research and acknowledge that the ASSURE nomogram is highly valuable for significant value for postoperative risk assessment in RCC, and its applicability extends to pRCC as well. The ASSURE nomogram was developed for predicting DFS and OS, and the risk factors in this nomogram are age, tumor size, histology, grade, coagulative necrosis, lymph node involvement, vascular invasion and sarcomatoid features. As the ASSURE cohort lacks molecular information and WSI data, our nomogram is not directly applicable to the ASSURE cohort. Thus, we applied the ASSURE nomogram to our training set and independent validation set (the TCGA set has no vascular invasion information). The results showed that, in the training set, the ASSURE nomogram predicted DFS and OS with a C-index of 0.650 and 0.649, respectively. These values are significantly lower than those of our nomogram (C-index: 0.774 and 0.758, respectively; $p < 0.001$). Similarly, in the independent validation set, the ASSURE nomogram predicted DFS and OS with C-indexes of 0.654 and 0.652, respectively, which are also lower than those achieved by our nomogram (C-index: 0.751 and 0.749, respectively; $p < 0.001$).

It is reasonable that our nomogram outperforms the ASSURE nomogram because our nomogram incorporates not only clinicopathological risk factors but also a molecular model and a WSI-based deep learning model, which enables our nomogram to identify the molecular and pathological characteristics of tumors more accurately.

5. The 2022 WHO recommendations emphasize the use of molecular subtyping of PRCC based on oncogenic drivers (see Lobo et al. Histopathology 2022, PMID: 35596618). For example, certain fusion partners of TFE3 confer a worse prognosis than others in MiT family renal cell carcinoma (which was often previously diagnosed as PRCC in the past). Similarly, SMARCB1-deficient renal cell carcinoma and fumarate hydratase-deficient renal cell carcinoma have worse prognosis than the KRAS+ papillary renal neoplasm with reverse polarity, while all three of these entities would have been classified as PRCC in the past, including in the TCGA. What is the added value of the proposed nomogram once these molecular drivers have been accounted for?

Our reply:

The issue raised here is worth exploring. The identification of molecular subtypes, including TFE3-rearranged RCC, fumarate hydratase-deficient RCC, and SMARCB1-deficient RCC, enhances the clinical evaluation of RCC prognosis. We think that an increasing number of molecular subtypes will be identified to assist RCC prognosis prediction in the future. Molecular subtypes that either have been identified or will be discovered in the future, along with our multi-classifier system can complement each

other rather than replace each other, to provide more accurate guidance for postoperative personalized treatment strategies going forward.

REVIEWERS' COMMENTS

Reviewer #1 (Remarks to the Author):

The authors have addressed my comments.

Reviewer #2 (Remarks to the Author):

The authors answered all my comments and added some additional text and analysis to address the issues that I highlighted.

One issue brought up by another reviewer and myself was that the high and low risk groups were defined internally for each data set. The authors provided a good explanation of why this was done, and this makes sense, but it does not address the actual issue.

That is when you want to run a test on a new single sample how do you know if a sample is in the high or low risk group? Do you run exactly the same tests as were performed in one of your data sets and use that to define whether the new sample falls in the high or low risk group? It is a fundamental issue with this type of classifier that it works by splitting a large group in two and then showing that one half is worst, but how does this help define the status of a new sample not in that group. This can influence the practicality of these tests. For instance, the authors mention the ClearCode34 mRNA-based signature analysis, and this has been confirmed in many cohorts and is now over a decade old. Yet, it has not, to my knowledge, been applied as an actual test in patients.

This does not need to be addressed within this paper but within the limitations section of the discussion there should be some comment on the issues surrounding implementing such tests. It should be stated that establishing defined criteria for the identification of each risk state dependent upon methodology would be beneficial in supporting the practical use of this classifier system to evaluate real patients.

I have no further comments in addition to the one stated above.

Reviewer #3 (Remarks to the Author):

The authors have sufficiently addressed my concerns.

Response to the reviewers' comments

Reviewer #1:

The authors have addressed my comments.

Reviewer #2:

1. The authors answered all my comments and added some additional text and analysis to address the issues that I highlighted..One issue brought up by another reviewer and myself was that the high and low risk groups were defined internally for each data set. The authors provided a good explanation of why this was done, and this makes sense, but it does not address the actual issue.

That is when you want to run a test on a new single sample how do you know if a sample is in the high or low risk group? Do you run exactly the same tests as were performed in one of your data sets and use that to define whether the new sample falls in the high or low risk group? It is a fundamental issue with this type of classifier that it works by splitting a large group in two and then showing that one half is worst, but how does this help define the status of a new sample not in that group. This can influence the practicality of these tests. For instance, the authors mention the ClearCode34 mRNA-based signature analysis, and this has been confirmed in many cohorts and is now over a decade old. Yet, it has not, to my knowledge, been applied as an actual test in patients.

This does not need to be addressed within this paper but within the limitations section of the discussion there should be some comment on the issues surrounding implementing such tests. It should be stated that establishing defined criteria for the identification of each risk state dependent upon methodology would be beneficial in supporting the practical use of this classifier system to evaluate real patients.

I have no further comments in addition to the one stated above.

Our reply:

Thank you once again for your valuable feedback. We added this limitation

concerning this issue (Line 339-343): Although using the median risk score as a cutoff is a common practice in many studies, this approach's population-specific nature restricts its clinical application. Therefore, this multi-classifier system requires further validation through prospective studies in large-scale, multi-center clinical trials that encompass additional geographic regions before it can be widely applied in clinical settings.

Reviewer #3:

The authors have sufficiently addressed my concerns.